

# Long-term Prediction of the Gulf Stream Meander Using OceanNet: a Principled Neural Operator-based Digital Twin

Michael Gray[1], Ashesh Chattopadhyay[2], Tianning Wu[1], Anna Lowe[1], and Ruoying He[1]

[1]North Carolina State University, Raleigh, North Carolina, 27695, United States
[2]University of California- Santa Cruz, Santa Cruz, California, 95060, United States

**Correspondence:** Michael Gray (magray@ncsu.edu)

**Abstract.** Many meteorological and oceanographic processes throughout the eastern United States and western Atlantic Ocean, such as storm tracks and shelf water transport, are influenced by the position and warm sea surface temperature of the Gulf Stream (GS)- the region's western boundary current. Due to highly nonlinear processes associated with the GS, predicting its meanders and frontal position have been long-standing challenges within the numerical modeling community. While the weather and climate modeling communities have begun to turn to data-driven machine learning frameworks to overcome analogous challenges, there has been less exploration of such models in oceanography. Using a new dataset from a high-resolution data-assimilative ocean reanalysis (1993-2022) for the Northwest Atlantic Ocean, OceanNet (a neural operator-based digital twin for regional oceans) was trained to identify and track the GS's frontal position over subseasonal-to-seasonal timescales. Here we present the architecture of OceanNet and the advantages it holds over other machine learning frameworks explored during development while demonstrating predictions of the Gulf Stream Meander are physically reasonable over at least a 60-day period and remain stable for longer.

## 1 Introduction

### 1.1 Background: The Gulf Stream Meander

The Gulf Stream (GS) is part of the Atlantic Ocean's Western Boundary Current. Easily identifiable by sea surface height (SSH) contours (Fig. 1), the GS carries warm equatorial water northward to the mid-to-high latitudes. The GS can be divided into two frequently studied segments: the Loop Current, which flows into and out of the Gulf of Mexico, and the Gulf Stream Meander (GSM), which extends to the east once the GS passes Cape Hatteras, North Carolina (Fig. 1). Due to its vast spatial coverage and anomalously warm temperatures, the GS influences much of the weather along the eastern coast of the United States as well as Western Europe (Minobe et al., 2008). Given its importance, there has been significant effort among modelers to forecast the position of the GS across various timescales.

Robinson et al. (1988) attempted to model a 26-day period of the "GSM and Ring region" using feature modeling techniques derived from remote sensing data and the Harvard Quasi-Geostrophic Open Boundary model. The authors carried out multiple experiments across durations, GSM positions, and combinations of sea surface temperature, topography, and boundary conditions being present. The following features were determined to be imperative to correctly simulate and achieve a convincing





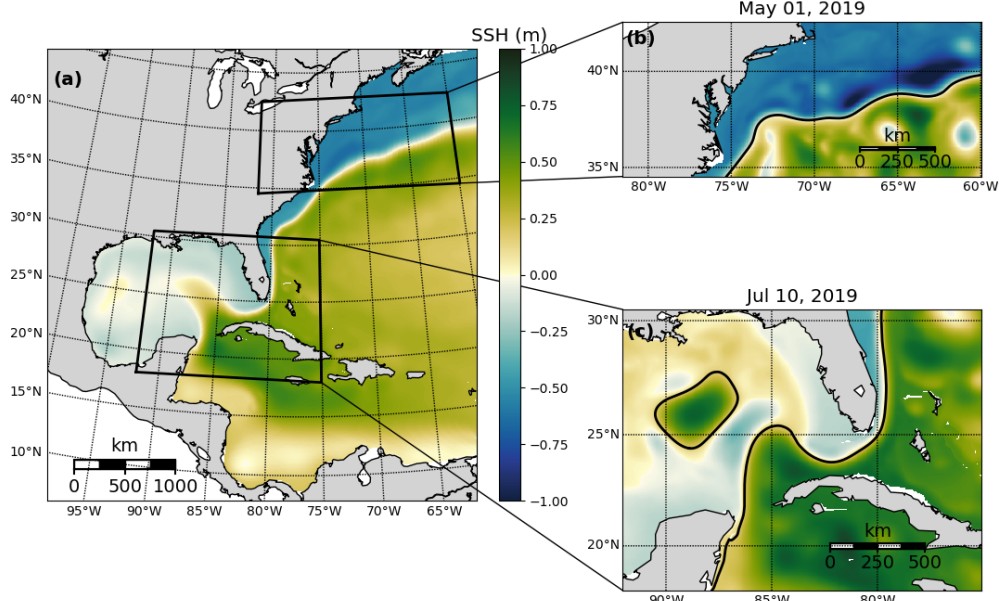

**Figure 1.** (a) The domain for the reanalysis data covering the northwestern Atlantic. The two subdomains used to develop OceanNet, specifically (b) the GS separation point and GSM from the central US east coast to 60°W and (c) the Loop Current eddy-shedding region in the eastern Gulf of Mexico, extending from 92°W into the Atlantic 75°W (not explored in this study). The mean SSH from from 1993-2020 in the reanalysis data is shown in (a), while (b) and (c) depict daily mean SSH on May 1, 2019, and July 10, 2019, respectively. All three domains share the same color scale.

GSM and Ring region: ring (eddy) formation via GSM breakoffs, ring coalescence with the GSM, and ring-ring mergers, interactions, and contacts. Chen et al. (2014) modeled a case study of one particular eddy: a warm core eddy event that lasted 27 days, detached and reattached the to GSM, and influenced heat and salt fluxes on the order of 6-9 times larger than mean values. While the authors do mention this was likely the largest and most energetic event in decades, visual inspection of sea surface variable plots (temperature, height, currents, etc) demonstrates how mesoscale eddy structures are frequently present
around the GSM. These structures, as seen in Chen et al. (2014), can greatly influence the overall circulation and fluxes of state variables.

It has been a challenging task for numerical models to robustly simulate and predict GS dynamics, especially its separation point off Cape Hatteras, North Carolina as well as mesoscale activity along the GSM (Chassignet and Marshall, 2008). A horizontal resolution of at least 1/10th degree is necessary to achieve a realistic separation point (Chassignet and Marshall,





2008). Higher resolutions are needed to properly represent the variabilities in GSM, including GS meanders and eddies, and the zonal penetration of the GS (Chassignet and Xu, 2017).

Since GSM simulations are the result of interactions on relatively small scales that propagate to much larger processes, numerical models must be carefully calibrated and include high-resolution physics. The open boundary to the east of the GS exacerbates these challenges by requiring larger domains to properly capture meridional fluxes into the system. These

compounding factors result in the need for massive compute power, time, and funding for numerical modelers. Conversely, machine learning predictions require a fraction of the resources; while these models may be slow to train, taking hours or even days, they result in extremely fast, cheap models that can exponentially reduce compute times and cost without making sacrifices in terms of resolution.

## 1.2    Machine Learning in Marine Sciences

In the weather and climate modeling communities, data-driven machine learning methods have become a popular field of exploration and have delivered promising results in the prediction of complex atmospheric circulation (Pathak et al., 2022; Bi et al., 2023b; Lam et al., 2023). Such models have demonstrated the aforementioned advantages of machine learning while outperforming state of the art numerical weather models for lead times of 8-10 days (Pathak et al., 2022; Bi et al., 2023b; Lam et al., 2023). A significant limitation of these models arises when they are integrated over longer time scales (two weeks or

longer), leading to instability and the emergence of nonphysical features (see Chattopadhyay and Hassanzadeh, 2023, Fig. 1)). The cause of this instability was identified as "spectral bias", an inductive bias in all deep neural networks that hinders their ability to capture small-scale features in turbulent flows. Chattopadhyay and Hassanzadeh (2023) proposed a potential solution in the form of a framework to construct long-term stable digital twins for atmospheric dynamics. That is not to say numerical weather models have a prediction horizon much longer (if at all) than two weeks, but the fact that machine learning atmospheric

models are as accurate as they are on a global scale for any lead time- even if they grow unstable- is a tremendous feat. While these advances are proving fruitful for the atmospheric sciences, there has yet to be progress of equivalent magnitude in predicting physical systems in marine sciences.

Efforts in emulating ocean dynamics with deep learning-based approaches have primarily focused on predicting large-scale circulation features such as those resolved by empirical orthogonal functions (EOFs) or on constructing low-dimensional

representations (Wang et al., 2019; Agarwal et al., 2021). Zeng et al. (2015) demonstrated the ability to predict SSH fields in the Gulf of Mexico associated with the Loop Current with about a four week lead time (up to six weeks in some cases) based on principle component time series of satellite-observed SSH fields. Zeng et al. (2015) later became the basis of Wang et al. (2019), where, after decomposing SSH fields in the Gulf of Mexico into principle component time series, the authors used a recurrent neural network, the Long Short Term Memory model (LSTM), to make a temporally-informed prediction.

This method led to impressive results claiming 12 weeks of predictability off a single input image of SSH, with persistence used as a baseline. While Zeng et al. (2015) and Wang et al. (2019) are both impressive, they share a similar problem of neglecting small-scale interactions that are important to the propagation of larger scale features such as the separation of a loop current eddy from the Loop Current. These studies are certainly a step in the right direction, but there has yet to be a global,



multivariate, data-driven ocean model similar to the weather models seen in Pathak et al. (2022), Bi et al. (2023b), and Lam
et al. (2023)

In an attempt to advance the progress of machine learning in marine sciences, examined here is the development and performance of a neural operator-based digital twin for the northwest Atlantic Ocean's western boundary current, named OceanNet, built upon the same principles as the FouRKS model introduced in Chattopadhyay and Hassanzadeh (2023). OceanNet relies on a Fourier neural operator (FNO), which incorporates a predictor-evaluate-corrector (PEC) integration scheme to suppress autoregressive error growth. Additionally, a spectral regularizer is employed to mitigate spectral bias at small scales. OceanNet is trained on historical SSH data from a high-resolution northwest Atlantic Ocean reanalysis and demonstrates remarkable stability and competitive skills. OceanNet, on average, outperforms SSH predictions made by the state-of-the-art Regional Ocean Modeling System (ROMS) across a 120-day period while maintaining a computational cost that is 4,000,000x cheaper (ROMS:10 hours across 144 CPUs; OceanNet: 1.18 seconds on a single NVIDIA A100 GPU).

This paper focuses on comparing variations of OceanNet and the arrival at the best architecture. For an in-depth discussion of the theory behind OceanNet and each of its components, see Chttopadhyay et al. (2023). For an investigation into the performance of OceanNet in the Gulf of Mexico, a highly-dynamic region with multiple mesoscale features, with a focus on the physical processes observed in the region, see Lowe et al. (2024).

## 2 Data & Methods

### 2.1 Northwest Atlantic Ocean Reanalysis

A high-resolution northwest Atlantic regional ocean reanalysis dataset was utilized to train OceanNet (Fig. 1)(He et al.; Wu and He). This reanalysis was generated using ROMS with the ensemble Kalman filter data assimilation method (EnKFDA). The dataset features a horizontal resolution of 1/25th degree with 50 vertical layers. For surface atmospheric forcing, data from the European Center for Medium-Range Weather Forecasting Reanalysis v5 (ERA5) was employed while open boundary conditions were derived from the Copernicus Global Ocean Physics Reanalysis (GLORYS). Ten major tidal constituents from the Oregon State University TPXO tide database were used. The model incorporated 120 river inputs, sourced from the National Water Model and climatological datasets. The temporal scope of the reanalysis data used spans from January 1, 1993 to December 31, 2020, with daily averaged output. The assimilated observations encompass a variety of sources, including AVHRR and MODIS Terra sea surface temperature, AVISO along-track sea surface height anomaly, glider temperature and salinity observations from the Integrated Ocean Observing System (IOOS), and the EN4 dataset which aggregates data from Argo floats, shipboard surveys, drifters, moorings, and other sources. The EnKFDA method used has no influence or retainment from future timesteps, as opposed to 3D and 4D-var methods, which include forward and backward passes over the results to ensure continuity. The resulting data assimilative ocean reanalysis allows the training of OceanNet with a time-space continuous SSH dataset and no external knowledge, as would be the case for observations.





## 2.2 Model development

### 2.3 Deep Learning Ocean Prediction

One of the groundbreaking machine learning models for weather prediction was introduced by Weyn et al. (2019), called the Deep Learning Weather Prediction (DLWP) model, and claimed the ability to predict 500-hPa geopotential height at forecast lead times of up to three days and can "...easily outperform persistence, climatology, and the dynamics-based isotropic vorticity model, but not beat an operational full-physics weather prediction model". Furthermore, Weyn et al. (2019) showed their DLWP model can output realistic atmospheric states for up to 14 days. The capabilities of the DLWP made it an attractive starting point for modeling the SSH field in regional oceans, and thus became the first iteration of OceanNet- henceforth referred as the Deep Learning Ocean Prediction model (DLOP).

DLOP is a relatively simple U-Net and overall simpler than DLWP, but the core idea is the same: pixel-wise connections of two-dimensional fields of physical variables between timesteps are sufficient to predict the evolution of such fields through time. The training of DLOP consisted of passing randomly shuffled SSH images from the reanalysis dataset from years 1993 through 2018 with a simple constraint of mean squared error of the predicted field. Due to the slower evolution of ocean states than that of the 500-hPa fields used in DLWP, a lead time of four days was used for training. The SSH fields were resampled to five-day running mean fields to remove the high-frequency noise associated with SSH such as tidal variations. The specifics of training of DLOP are almost identical to the final training of OceanNet- explained more thoroughly in Sect. 2.5.

Once trained, DLOP was used to autoregressively predict SSH fields out to 120 days. Initially, predictions from DLOP appeared to be performing somewhat well- Root-Mean-Squared Error (RMSE) and Correlation Coefficients (CC) for a four or eight day prediction were on-par or better than other predictive models (timeseries of metrics for models discussed here can be found in Sect. 3); however, predictions using DLOP tended to a mean state within a couple of timesteps before eventually growing completely unstable and thus nonphysical (Fig. 2). A thorough investigation of DLOP led to a similar conclusion to that of Chattopadhyay and Hassanzadeh (2023); the shortcomings of the DLOP's U-NET backbone reside in its inability to capture small-scale features in turbulent flows evident by the mismatch of high wavenumbers present in the fields.

Efforts were made to try and combat this documented phenomenon, specifically those prescribed by Chattopadhyay and Hassanzadeh (2023) known as the FouRKS framework. The FouRKS framework consists of employing numerical integration (Sect. 2.4.2) and spectral regularization (Sect. 2.5.1) techniques to suppress autoregressive error growth and encourage the model's attention to correctly predict the smaller scale features present. The inclusion of numerical integration in the model architecture did improve the stability horizon of the model and thus led to much lower metric values of RMSE and CC over the 120 day prediction period, but an investigation into the actual images being produced by the model showed that DLOP was slowly tending to what can only be described as a background state of the GSM. While the numerical integration techniques helped to some degree, the spectrum of the model was still an inaccuracy of interest that could potentially be fixed when paired with the spectral regularizer. Unfortunately, both with and without the presence of the numerical integrator, the spectral regularizer caused DLOP to become even more unstable than before, as can be seen from the metrics alone- after two timesteps, the model loses all physicality and immediately propagates noise throughout the domain.





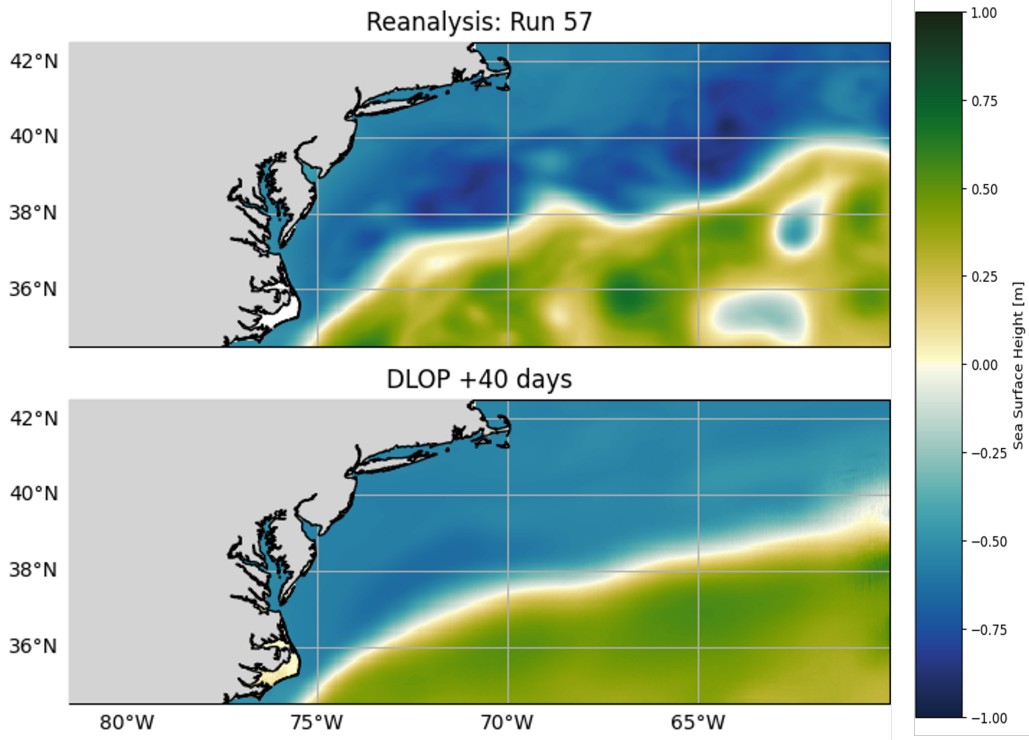

**Figure 2.** Prediction performance of a DLOP on the GSM region at 40 days. (Top) The SSH field from the reanalysis dataset 40 days after DLOP's initialization. "Run 57" refers to the (randomly selected) 57th initialization of DLOP as described in Sect. 3. (Bottom) The SSH field of DLOP's 40th day of prediction, demonstrating the model tending to the background state of the GSM.

The failures of DLOP quickly evolved into a complicated problem. Despite the documentation for what was being seen by

Chattopadhyay and Hassanzadeh (2023), a stable and accurate U-Net for ocean prediction could not be achieved. In numerical modeling, the suppression of error can typically be handled by integrating on shorter time scales or by adding constraints to the system to discourage the development of instability. For DLOP, integrating on shorter time scales would mean predicting with a smaller lead time each time step, which was attempted. Though the results are not shown here, extensive trial and error revealed that integrating on lead times less than four days led to DLOP not propagating anything forward- in other words, the

evolution of the SSH field over three days or fewer is so small, the training process resulted in DLOP determining it would achieve the lowest error if it kept the field completely static with every prediction. This may not be a problem with atmospheric prediction, as in DLWP, because the evolution of the atmosphere is noticeable over much shorter timescales. As for constraining the system to suppress error, this was the intent of the spectral regularization techniques but no improvement was observed.





## 2.4 OceanNet

Given the analysis of the DLOP results is consistent with previous literature (see Sect. 2.3) and that there was a noticeable improvement in the stability of DLOP with the inclusion of the numerical integrator, the techniques seen in Chattopadhyay and Hassanzadeh (2023) continued to be employed. While the spectral regularization scheme did not provide much (if any) improvement to DLOP, the idea of constraining the system's distribution of wavenumbers and stabilizing autoregressive prediction remained attractive; however, instead of using the typical two-dimensional convolutions with a U-Net structure, Fourier

Neural Operators (FNOs) with a multi-timestep loss function were thought to provide similar behavior. This section provides more information regarding FNOs and numerical integration while Sect. 2.5 further explains the spectral regularization and the multi-timestep constraints used in the final version of OceanNet.

### 2.4.1 Fourier Neural Operator (FNO)

OceanNet is built upon the FNO (Li et al., 2020). FNOs behave in a similar manner to typical two-dimensional convolutions in

a U-Net, but in Fourier space: a Fourier transform is performed on the input data, the highest Fourier modes are reduced to zero, then an inverse Fourier transform brings the data back to a real space where it is concatenated with the input image (Fig.3a). FNOs were introduced in  Li et al. (2020) where the authors demonstrate higher performance benchmarks in terms of speed and error than any other deep learning technique to date when modeling complex fluid flow problems such as the Burger's Equation, Darcy Flow, and the Navier-Stokes equations. Due to performing operations in Fourier space, the FNO is considered

to be resolution agnostic- which is an advancement in and of itself since prior methods of deep learning for image-to-image translation required consistent use of the training data's resolution. The speed of FNOs comes from the various advancements in the computer science fields which have led to extremely efficient implementations of the Fast Fourier Transform; furthermore, FNOs do not rely on scanning the information in two-dimensional space as convolution and pooling layers do and instead are integrating the whole field at once.

As with DLOP, training utilizes labeled pairs of historical five-day-running mean SSH data in the GSM, $\mathbf{X}(t)$ (image), $\mathbf{X}(t+\Delta t)$ (label), and $\mathbf{X}(t+2\Delta t)$ (label), where $\Delta t = 4$ days. The training assumes the governing partial differential equation for the reduced ocean system involves ocean states $\mathbf{X}(t)$:

$$\frac{d\mathbf{X}(t)}{dt} = \mathbf{F}\left(\mathbf{X}\left(t\right)\right). \tag{1}$$

To integrate the system from the initial condition, $\mathbf{X}(t)$, Eq. (1) is represented in its discrete form:

$$\mathbf{X}(t+\Delta t) = \underbrace{\mathbf{X}(t) + \int_{t}^{t+\Delta t} \underbrace{\mathbf{F}\left(\mathbf{X}\left(t\right)\right) dt}_{\mathcal{N}[\circ,\theta]}}_{\mathbf{H}[\circ]}. \tag{2}$$



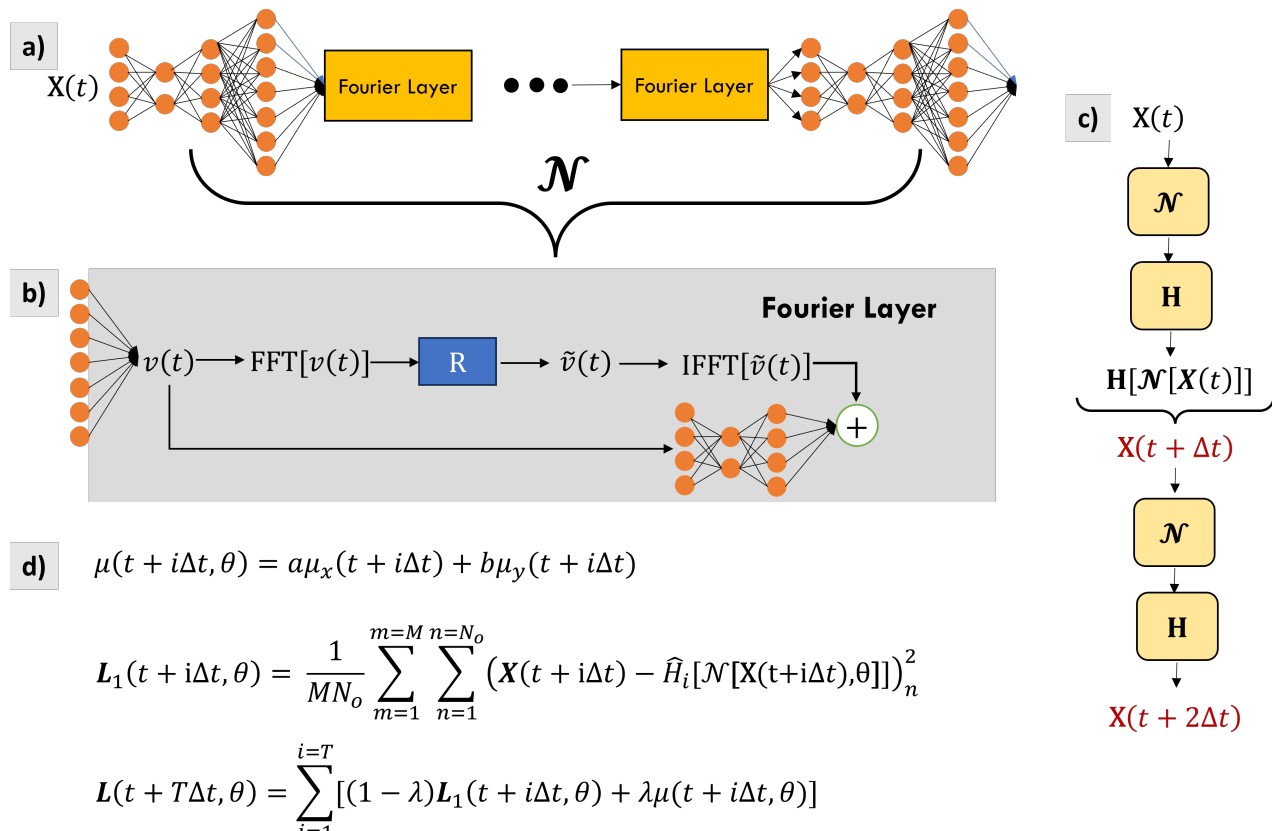

**Figure 3.** (a) A schematic of the OceanNet model with input image $\mathbf{X}(t)$. (b) The Fourier Neural Operator, depicted as $\mathcal{N}$ where $\nu(t)$ represents some two-dimensional field. (c) two-time-step training scheme with numerical integrator $\mathbf{H}$. (d) The loss function used, constructed by the spectral regularizer $\mu$ and MSE $\mathbf{L}_1$ for $M$ samples to give the generalized multi-time-step function $\mathbf{L}$.

Here, $\mathcal{N}[\circ,\theta]$ is the neural network which parameterizes $\mathbf{F}$ with four Fourier layers, similar to Li et al. (Li et al., 2020), each layer retaining $64$ Fourier modes. $\theta$ represents the $\approx 80 \times 10^6$ trainable parameters of the FNO. $\mathbf{H}[\circ]$ represents some implicit integration scheme, e.g. the final version of OceanNet uses a higher-order predictor–evaluate–corrector (PEC) integrator (Sect. 2.4.2).

$\rightarrow \mathbf{X}(t + \Delta t) = \mathbf{H}[\mathcal{N}[\mathbf{X}(t),\theta]]$                                                    (3)

Before officially characterizing the issues seen in DLOP as a problem in predicting the proper distribution of features of various wavenumbers and focusing on resolving this disparity alone, DLOP was replaced with an FNO and was expanded to include depth-averaged currents so the conservation of potential vorticity (PV) and geostrophy (GEO) could be examined. This analysis was conducted in the Gulf of Mexico as opposed to the GS (model regions were explored simultaneously, see Fig. 1 for

an example of the Gulf of Mexico domain), but gives great insight to the level of conservation of PV and GEO due to the larger



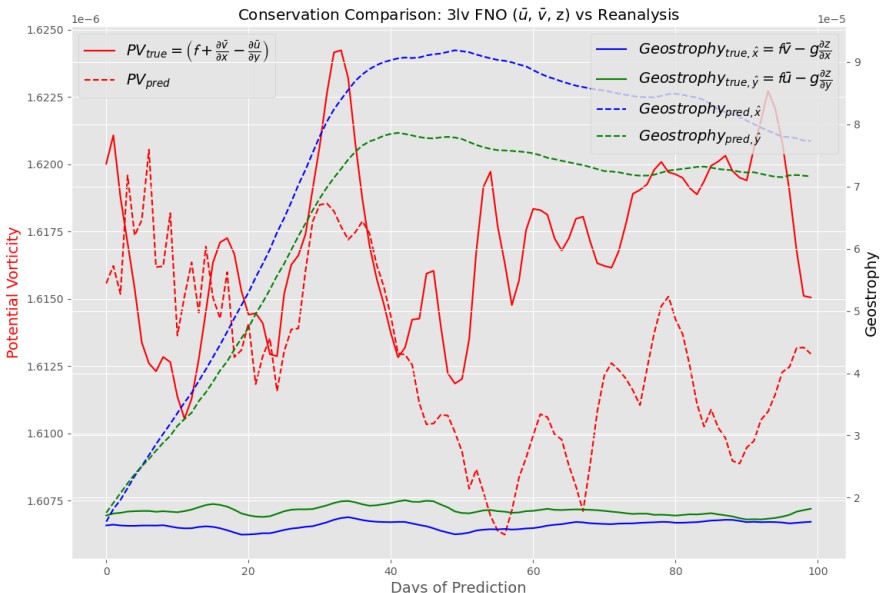

**Figure 4.** Comparison of the potential vorticity and the residual geostrophy in the FNO for the Gulf of Mexico. Note the scale of the GEO axis is 1e-5 and the scale of the PV axis is 1e-6.

proportion of the area covered by the mesoscale eddies present (Fig. 4). The residual GEO is close to zero in the reanalysis dataset, as expected, and the PV present remains around $1.1675 \mathrm{s}^{-1}$ for all 100 days used for comparison. The predicted SSH fields in this model run consistently have more residual GEO at all time steps, but on the order of $1E-5\mathrm{s}^{-1}$. In terms of PV, the model agrees quite well with the reanalysis until day 40, where the PV in the model is consistently less than that of the reanalysis, but on the order of $1E-3\mathrm{s}^{-1}$. Despite having no direct knowledge of conservation laws, the FNO appears to inherently conserve both PV and GEO on orders similar to that of the reanalysis dataset it was trained on. This iteration of OceanNet was not pursued further due to it suffering from the same spectral problems observed in the DLOP model (Sect. 2.3).

### 2.4.2 Predictor-Evaluate-Corrector (PEC) Integration Scheme

Similar to the higher-order integration scheme in the form of fourth-order Runge-Kutta (RK4) used in Chattopadhyay and Hassanzadeh (2023), the PEC scheme is implemented in OceanNet, represented by the operator, $\mathbf{H}[\circ]$. The operations in $\mathbf{H}$ are given by:

$$\mathcal{N}_1 = \mathcal{N}\left[\mathbf{X}(t), \theta\right], \tag{4a}$$

$$\mathbf{H}\left[\mathcal{N}\left[\mathbf{X}(t), \theta\right]\right] = \mathbf{X}(t) + \mathcal{N}\left[\mathbf{X}(t) + \frac{1}{2}\mathcal{N}_1, \theta\right]. \tag{4b}$$



The final predicted state is given by $\mathbf{H}\left[\mathcal{N}\left[\mathbf{X}\left(t\right),\theta\right]\right]$.

Although most of the higher-order integration schemes, including RK4, demonstrate good performance for this problem, PEC has been identified as the most effective choice due to its compromise between higher-order integration and memory consumption during training. A theoretical study of the effect of each integration scheme on the inductive bias of the trained $\mathcal{N}$ is an active area of research, especially for understanding the role it plays on the subsequent spectral bias (Krishnapriyan et al., 2022).

## 2.5   Training and Validation

OceanNet for the GSM was trained on five-day-running mean SSH reanalysis fields (to remove high-frequency features such as tides) for the years of 1993 through 2018. The years of 2019 and 2020 were reserved for validation and testing. Before training, all of the data was randomly shuffled. There are two general steps to training OceanNet: single-timestep training and multi-timestep training. Prior to either training segment, the SSH data is normalized by removing the pixelwise 30-year mean and dividing by the pixelwise 30-year standard deviation. After normalization, the input images are fed into the model where a four day lead prediction is given. For single-timestep training, the prediction and the reanalysis image of the corresponding day are evaluated by the loss function (described in Sect. 2.5.1). For multi-timestep training, the output of the model is fed back through the model to produce an eight day lead prediction which is then evaluated by the two-timestep loss function. Based on hyper-parameter optimization via the Optuna python package, the optimal training workflow included 180 epochs of single-timestep training followed by 180 epochs of multi-timestep training. Two values were used to validate OceanNet's performance: the Modified Hausdorff Distance (explained in Sect. 3) and the value of the loss function.

### 2.5.1   Spectral Regularization in Fourier Space and the 2-timestep Loss Function

In OceanNet's loss function, spectral regularization was incorporated based on Fourier transforms, introduced in Chattopadhyay and Hassanzadeh (2023). This is in addition to the standard mean squared error loss (MSE) function, which is computed exclusively for grid points located over the ocean.

The spectral regularizer penalizes deviations in the Fourier modes present in the SSH field at small wavenumbers. Such deviations arise due to spectral bias, which represents an inherent inductive bias in deep neural networks (Xu et al., 2019; Chattopadhyay and Hassanzadeh, 2023). This bias is responsible for their limitations in learning the fine-scale dynamics of turbulent flow. This regularization was carried out across both x and y dimensions to ensure that the high wavenumbers in the Fourier spectrum of SSH remain consistent with the target Fourier spectrum.

$$\mu_x\left(t+\Delta t,\theta\right)=\frac{1}{M(K_{Nx}-K_{cx})}\sum_{m=1}^{m=M}\sum_{k=K_{cx}}^{k=K_{Nx}}\left|\widehat{\mathcal{F}}_x\left[\mathbf{X}\left(t+\Delta t\right)\right]-\widehat{\mathcal{F}}_x\left[\mathbf{H}\left[\mathcal{N}\left[\mathbf{X}\left(t\right),\theta\right]\right]\right]\right|_k. \tag{5a}$$

$$\mu_y\left(t+\Delta t,\theta\right)=\frac{1}{M(K_{Ny}-K_{cy})}\sum_{m=1}^{m=M}\sum_{k=K_{cy}}^{k=K_{Ny}}\left|\widehat{\mathcal{F}}_y\left[\mathbf{X}\left(t+\Delta t\right)\right]-\widehat{\mathcal{F}}_y\left[\mathbf{H}\left[\mathcal{N}\left[\mathbf{X}\left(t\right),\theta\right]\right]\right]\right|_k. \tag{5b}$$





$$\mu(t+\Delta t,\theta) = a\mu_x(t+\Delta t,\theta) + b\mu_y(t+\Delta t,\theta) \tag{6}$$

Here, $M$ is the number of training samples (batch size), $k$ represents a single Fourier mode, $K_N$ is the highest Fourier mode present along the respective axis, and $K_c$ is the "cutoff" Fourier mode i.e. the minimum mode of interest. After extensive trial and error, the best performance of OceanNet was observed with $K_{cx} = 10$ and $K_{cy} = 30$. $a$ and $b$ are scaling factors used to ensure the order of magnitude of $\mu$ agrees with the order of magnitude of MSE (Eq. 7), as can be seen in the full loss function for $t + \Delta t$ given by $\mathbf{L}(t+\Delta t,\theta)$:

$$230 \quad \mathbf{L}_1(t+\Delta t,\theta) = \frac{1}{MN_o}\sum_{m=1}^{m=M}\sum_{n=1}^{n=N_o} (\mathbf{X}(t+\Delta t) - \mathbf{H}[\mathcal{N}[\mathbf{X}(t),\theta]])_n^2 \tag{7}$$

$$\mathbf{L}(t+\Delta t,\theta) = (1-\lambda)\mathbf{L}_1(t+\Delta t,\theta) + \lambda\mu(t+\Delta t,\theta) \tag{8}$$

where $\mathbf{L}_1$ is MSE over $N_o$ ocean grid points and $\lambda$ is a weighting factor.

During single-time-step training, a weighted loss function of spectral regularization and MSE is used to constrain the model. To stabilize the model over multiple autoregressive predictions, the loss function is generalized to incorporate the sum of the loss function evaluated at each predictive step. The number of time steps over which the loss is calculated can be extended to any number of autoregressive steps; however, with each increase in the number of time steps the memory requirement for the subsequent backpropagation process during training grows exponentially thus the compromise of two timesteps was reached.

## 3 Results

This section presents a comparison of mesoscale ocean circulation dynamics represented by spatio-temporal evolution of SSH fields generated by various iterations of DLOP and OceanNet with the dynamical ROMS forecast using independent reanalysis data. To assess the performance rigorously, both qualitative and quantitative measures are employed. The metrics for evaluating predictive accuracy of SSH include RMSE and CC, which are widely recognized and employed in forecasting (Pathak et al., 2022; Bi et al., 2023a; Lam et al., 2023; Chattopadhyay and Hassanzadeh, 2023). In addition, a specialized object-tracking metric to evaluate the prediction of major ocean features delineated by SSH contours is incorporated: the modified Hausdorff distance (MHD,Dukhovskoy et al. (2015)). MHD quantifies the comparison of predicted objects to their counterparts between grids; identical shapes at identical locations yield an MHD of zero. To calculate MHD, at least one shape needs to be identified in each image. For the GSM, the contour identifying the northern frontal boundary of the meander was chosen to be used in MHD calculations. This boundary of the GSM is indicated by using a contouring threshold of SSH pertaining to the average SSH across all points in the reanalysis dataset with geostrophic speeds exceeding the average zonal maximum. This SSH contour is approximately -0.17m. While this defined northern boundary of the GSM is not illustrated by a contour line on most figures, an example of it can be seen in Fig. 1(b). The choice of this method for defining the GSM's position proves convenient



since it provides a single object which is present in all images- if a contouring level which captures the shapes of individual eddies independent of the GSM where chosen, the calculation of the MHD score becomes tedious due to the possibility of having a mismatch in the number of objects between prediction and validation images. To provide a comprehensive assessment,

qualitative snapshots of the predicted SSH fields generated by OceanNet, ROMS dynamical forecasts, and the independent SSH fields derived from the reanalysis are shown.

The ROMS forecasts used for comparison consists of 69 uncoupled 120-day predictions initialized five days apart. For fair comparison with the reanalysis dataset, the five-day mean SSH fields of the ROMS output were compared. Since OceanNet has no knowledge of the atmospheric states nor ocean boundary conditions during its inferencing, the ROMS forecasts were forced

with persistent atmospheric and ocean boundary conditions for each run. This configuration of ROMS serves as a realistic use-case; in the absence of atmospheric forecasts, an operational version of ROMS would use persistent forecasting. An example of the 2 m air temperature and 10 m wind vectors used to initialize and force a single ROMS simulation is provided (Fig. 5).

A qualitative assessment reveals that OceanNet effectively captures the SSH propagation of undulations in the northern boundary of the GSM (Fig. 6). Moreover, OceanNet skillfully captures large-scale eddies traveling into and out of the domain,

even without receiving any boundary information. In contrast, ROMS dynamical model forecast tends to overpredict SSH and the meridional amplitude of the northern boundary. While it is sensitive to initial conditions, OceanNet remains physically consistent over long-term forecasts in this region. Also, OceanNet provides stable and physically reasonable SSH predictions for the GSM for at least 120 days (not shown for brevity).

For quantitative comparisons, predictions from each model (OceanNet, ROMS) and persistence are compared to the reanal-

ysis dataset to derive metrics at each day of prediction and are presented as averages of the $n^{th}$ day of prediction (Fig. 7). This method allows performance to be evaluated by forecast lead time across various initialization states; an evaluation of RMSE on the 20th day of prediction is an average measure of model performance with a forecast lead of 20 days given 69 different initial conditions. Each model was also compared to the saturation value of each metric: the 95th-percentile of the corresponding metric calculated from 1,000 random pairs of images from the entire reanalysis dataset (DelSole, 2004; DALCHER and

KALNAY, 1987). If a metric exceeds the corresponding saturation value, the confidence in the prediction is considered to be no more trustworthy that that of selecting a random field of SSH from the reanalysis dataset. It is also important to note that not just the means of the ensembled metrics are investigated, but the corresponding standard deviations of each metric are also considered. If the means of two objects of comparison fall within the standard deviations of each other, not much weight can be put into claiming one model to perform better than the other. In these manners, OceanNet consistently outperforms

ROMS in RMSE, CC, and MHD computed between the predicted SSH values and the reanalysis SSH over 120 days (Fig. 8). Persistence forecasting also fares reasonably well in this region due to the strong background state of the GSM; however, OceanNet can still outperforms persistence in all three metrics over 120 days on average. The MHD of OceanNet is shown to cross the saturation value of 62.34 km on day 60, suggesting the northern boundary of the GSM predicted by OceanNet is no better than selecting a random image from the reanalysis dataset. This is not to say that the position of the entire GSM is off,

but that the undulations present in the GSM's northern boundary are completely out of phase. When this is the case, OceanNet does maintain the correct relative position of the GSM while ROMS frequently places the GSM too far north or south.





**Figure 5.** An example of atmospheric conditions used to force the uncoupled numerical simulations in ROMS- used in the simulation initialized on March 7th, 2020. Variables shown: 2 m air temperature (shading) and 10 m wind vectors (every eighth grid point plotted for visual clarity).



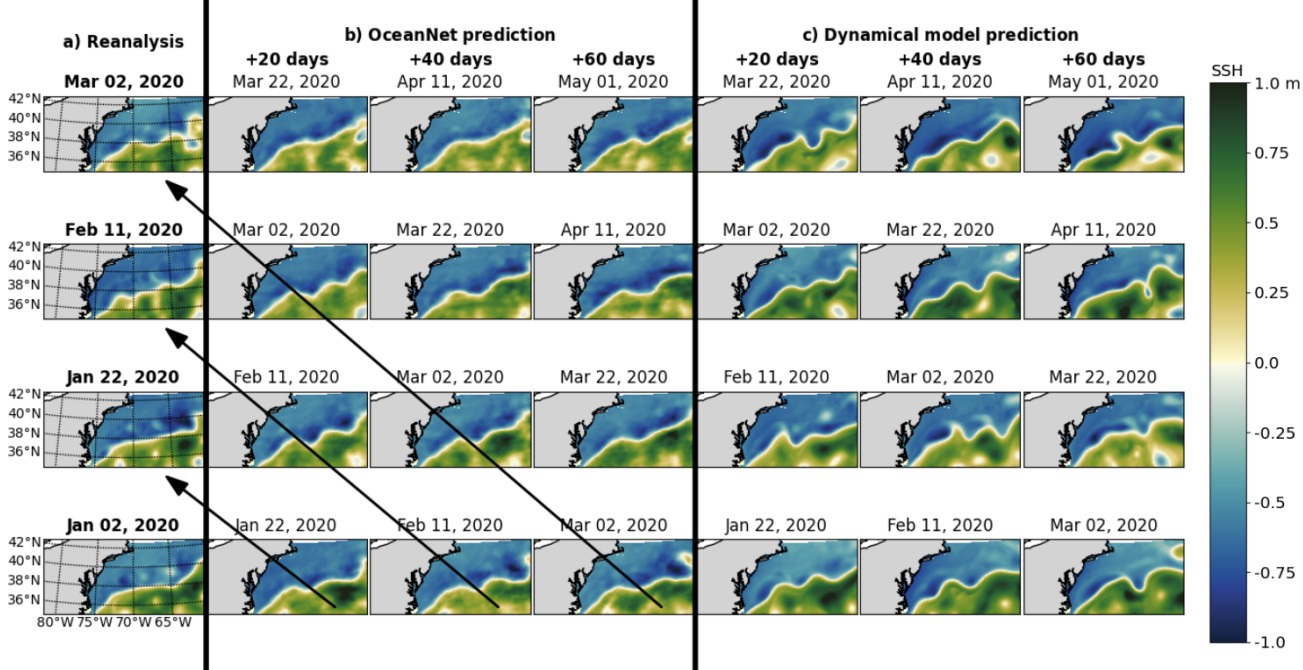

**Figure 6.** Performance of OceanNet for GS prediction. (a) SSH fields from the ocean reanalysis. (b) Predicted SSH generated by OceanNet. (c) ROMS dynamical model forecasts. In both OceanNet and dynamical model predictions, each row was initialized with the corresponding reanalysis data in the left column. SSH forecasts are provided for 20, 40, and 60 days. To evaluate the predictions, we can perform a diagonal comparison with the reanalysis SSH, as indicated by the black arrows in (b). The same diagonal comparison can also be conducted with the ocean reanalysis data for (c).

The RMSE, anomaly correlation coefficient (ACC), and MHD are compared between various iterations of DLOP and FNO models pertaining to integration schemes and loss function terms. The two integration schemes compared were the lack there of and PEC. The loss function terms compared were MSE and MSE with spectral regularization. The possible combinations of model type, integration schemes, and loss function terms yield eight models to compare, in the same manner as before (ensembled metrics, Fig. 7), between each other as well as with ROMS and persistence predictions.

Since RMSE is an indicator of the magnitude of values present, it is fair to say that RMSE is a measure of accuracy as well as stability; if RMSE is high, the magnitudes of the field being analyzed are, on average, unrealistic and if the RMSE continues to grow through time, the model is regarded as unstable. In terms of RMSE, the two DLOP models with spectral regularization included in the loss functions can immediately be identified as becoming unrealistic and unstable within a couple timesteps since they almost immediately cross the saturation threshold and continue to rise (Fig 9). The two DLOP models with only MSE in their loss function appear to perform well, especially when the PEC integrator is present, but the very basic DLOP model with MSE loss and no other augments does appear to become unstable around day 100. Out of all the DLOP models, the only one with competitive RMSE at all time steps is the iteration with PEC integration and MSE loss. For the FNO model,





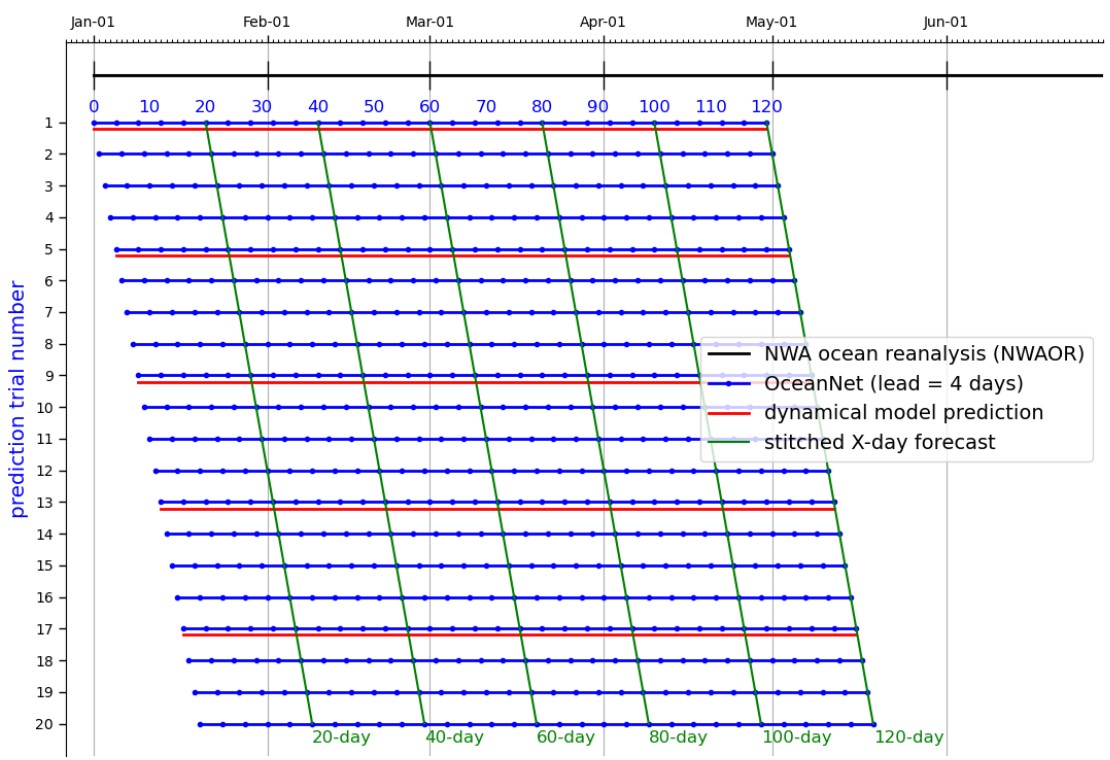

**Figure 7.** Visual explanation (example) of the ensembling of metrics for the evaluation of OceanNet across various ocean states. The explanation is framed with reference to OceanNet, but applies to the analysis of all iterations of DLOP and FNO models. The black line shows the time period covered by the reanalysis dataset with daily output. Blue lines represent individual OceanNet predictions trials spanning 120 days, each initialized one day apart, while blue dots indicate output timesteps of the model (every four days). Red lines represent individual ROMS predictions, which were initialized every five days with output given every day. Trials where the initialization dates between both ROMS and OceanNet align were compared qualitatively (visual comparison of output fields) along the green lines and quantitatively (RMSE, CC, and MHD) at all timesteps.

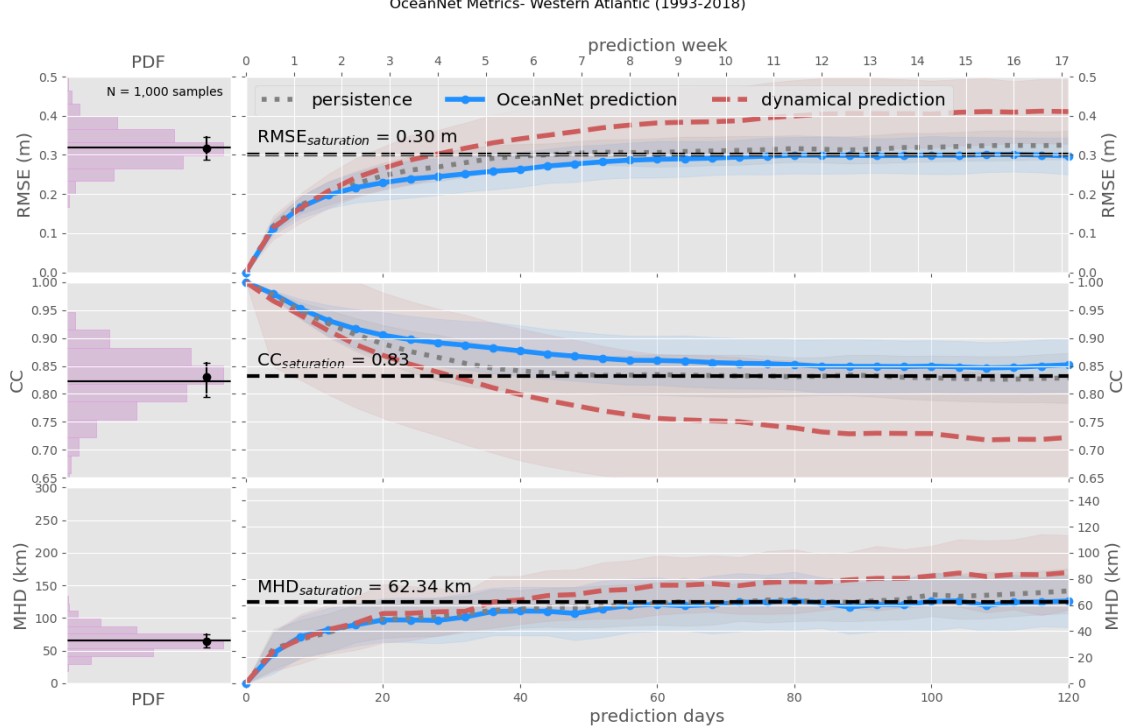

**Figure 8.** OceanNet's performance metrics in the northwest Atlantic: RMSE (top), CC (middle), and MHD (bottom), compared to the persistence forecast and ROMS dynamical model forecast. In the left column, probability density functions are presented, derived from 1,000 random pairs sourced from the training data spanning 1993 to 2018 (means shown by black horizontal lines). The performance statistics, calculated based on forecasts of 0-120 days, are displayed as mean values (lines) with standard deviations (shading). The black horizontal dashed lines denote saturation values, which are determined as 95% of the means derived from the random pairs. These representations illustrate how each method's statistics compare with the target SSH from the reanalysis dataset

all combinations of integration schemes and loss function terms are approximately the same; however, they all show slightly higher RMSE at day 60, that continues to increase in time, than the DLOP model with PEC integration and MSE loss.

The plots regarding RMSE are a great initial impression of performance, but other metrics are important to consider when choosing the best model. That said, the MHD plots tell a similar story to RMSE, with the only differences to note being: DLOP with MSE loss does not become unstable in terms of MHD, and all the FNO models remain under the saturation value for longer than the two best DLOP models identified in the RMSE plots (Fig. 10). Taking the analyses of RMSE and MHD together, it seems as though the best model may be any of the FNOs or DLOP with MSE loss. The final metric investigated is the ACC. Like the CC, ACC is a comparison of how closely correlated two sets of data are, but ACC considers the field with the long-term pointwise mean removed prior to comparison. The removal of the long-term mean allows comparison of the two datasets on a finer scale. From the ACC, the results of the RMSE and MHD analyses are confirmed and the qualifying best





**Figure 9.** Comparison of RMSE between combinations of models, integration schemes, and loss function terms. Values with RMSE at day 60 are indicated. Line colors and styles are indicated by the legend and represent the mean value across all ensemble members. Shading represents the range of +/- 1 standard deviation. (a-d) DLOP models. (e-h) FNO models. (a,e) MSE loss function with no integration scheme. (b,f) MSE loss function with PEC integration. (c,g) MSE with spectral regularization and no integration scheme. (d,h) MSE with spectral regularization and PEC integration.



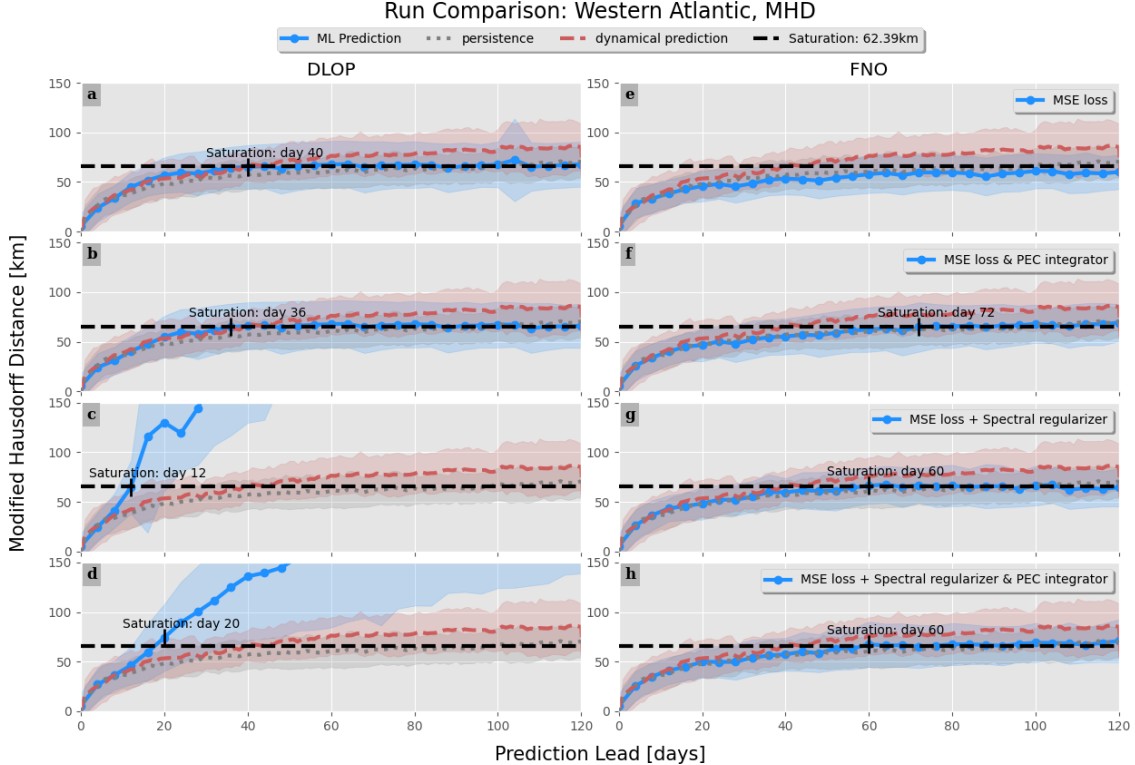

**Figure 10.** Same as Fig.9 but for MHD. The day at which each iteration of the models crosses the saturation value is indicated.

models are selected to be any version of the FNO and the DLOP model with PEC integration and MSE loss since these models
are at least competitive with the ROMS predictions across all timesteps in all three metrics (Fig. 11). This is the extent of the
analysis possible from the provided metrics so to identify the absolute best model one must compare the actual fields of SSH
predictions produced by each model to ensure they make physical sense.

While there are four versions of the FNO model which, metrically, appear to be competitive, extensive hyperparameter
tuning and subsequent verification revealed the best of these to be the FNO with PEC integration and MSE loss with spectral
regularization. This model, with the addition of the two-timestep loss described in Sect.2.5, became what is presented here
as OceanNet. Covering the individual results of hyperparameter tuning each FNO model and presenting then comparing the
verification of their physical fields is beyond the scopes of this paper. An example prediction of a single instance of prediction
with a lead of 40 days made by the best DLOP model, ROMS, and the finalized OceanNet model is shown to demonstrate the
difference between the physical fields predicted by each type of model (Fig. 12).





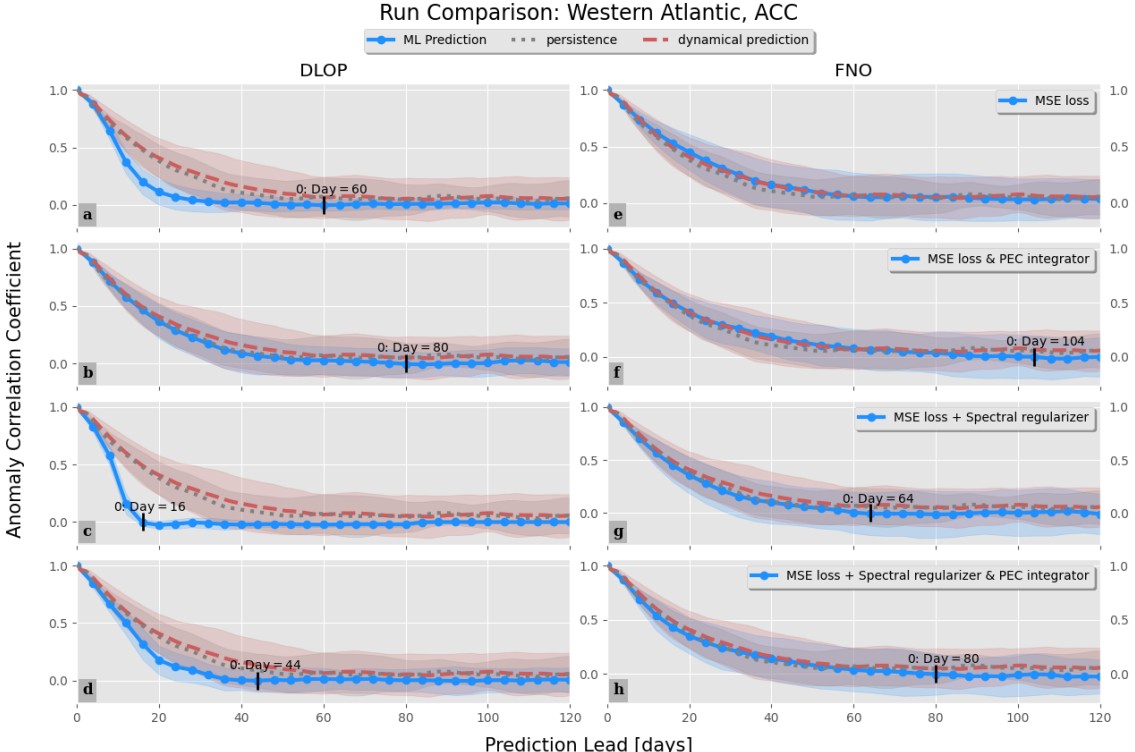

**Figure 11.** Same as Fig.9 but for ACC. The day at which each iteration of the models reaches a value of 0 is indicated.

## 4 DISCUSSION & CONCLUSIONS

This study demonstrates the capabilities of the neural operator-based OceanNet: a data-driven machine learning model for GSM prediction over subseasonal-to-seasonal time scales. The techniques explained throughout Sect. 2.2 (FNO, PEC-integration, spectral regularization, and multi-timestep criterion) mitigate autoregressive error growth and the spectral-bias seen in other
data-driven architectures, making OceanNet a solid candidate to function as a digital twin for long-term regional ocean circulation simulations.

Using high-resolution SSH data for the years 1993 through 2018, OceanNet was trained to predict ocean states with a four day lead time. The ability of OceanNet to autoregressively forecast the mesoscale ocean processes of the GSM over 60-120 days was evaluated by standard metrics used in machine learning and oceanographic communities. The results of this study
provide two main conclusions: OceanNet remains remarkably stable over many iterations of autoregressive prediction and the model consistently outperforms ROMS dynamical forecasting across various initial ocean states in terms of RMSE, CC, and MHD. In addition, an inherit advantage of machine learning models in general are their ability to inference at tremendous speeds (4,000,000 times faster, in this case). These results demonstrate the potential of utilizing scientific machine learning to





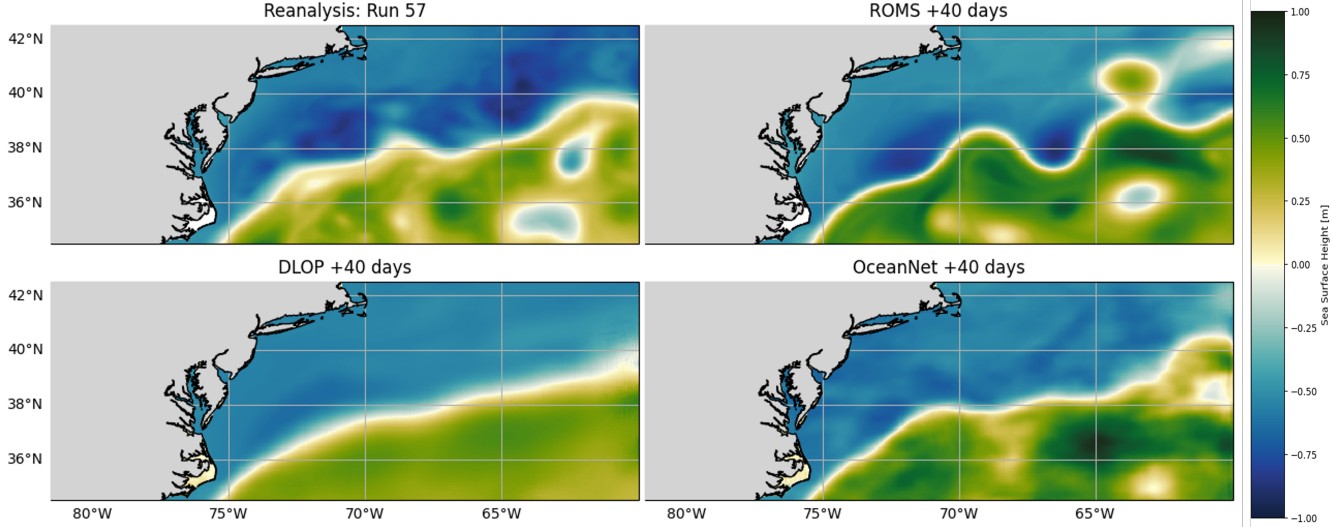

**Figure 12.** Prediction performance of a DLOP, ROMS, and OceanNet on the GSM region at 40 days. The SSH field 40 days after model initialization of "Run 57," referring to the 57th ensemble member (randomly selected) described in Sect. 3, for (Top-left) the reanalysis dataset, (Top-right) ROMS, (Bottom-left) DLOP, and (Bottom-right) OceanNet.

develop long-term stable, and accurate data-driven ocean models of great computational efficiency, paving the way for realizing
a data-driven digital twin encompassing the entire climate system.

While the skill of OceanNet is impressive, the conclusions presented here are not without limitations. This study was conducted and trained on a single ocean feature, of a single spatial and temporal scale, from a reanalysis dataset utilizing only a single variable. Real-world ocean forecasting systems operate with full-physics dynamical ocean models and real-time observational ocean data, covering dynamic processes across diverse spatial and temporal scales. The disparities between these
data sources and scales necessitate further investigation into OceanNet's performance across various ocean applications. The comparisons between OceanNet and ROMS can also be considered to have a major caveat: ROMS as a regional ocean model depends on persistent forcing conditions at open boundaries, for which persistent boundaries were provided in this study. While this method of modeling is operational in the sense that the absence of subseasonal-to-seasonal prediction often lacks forcing information on similar timescales thus persistence must be used, it may be more fair to compare the performance of OceanNet
to a model which does not require boundary conditions, such as a global ocean model. The use of a global model would be expensive due to the resolution OceanNet uses, so perhaps the best comparison could be done once OceanNet is expanded to cover the global ocean as well. In the meantime, the potential for OceanNet to include multiple state variables, such as surface currents, temperature, or even depth-averaged variables, could improve prediction of smaller scale circulations and events, such as shelf break jets and frontal currents as well as provide more variables to compare to numerical methods. In addition, Ocean-
Net produces very smooth, continuous fields which can potentially lead to an underestimation of the magnitudes of extreme





ocean events; therefore, additional research is imperative to assess OceanNet's performance under extreme ocean conditions, e.g., during severe storms.

Significant opportunities exist for improvement in both AI-based methods and dynamical model-based ocean forecasting. In the AI domain, potential advancements involve the integration of subsurface ocean states and additional ocean variables, 355 the incorporation of temporal dimensions through the training of four-dimensional deep networks, and the exploration of more complex network architectures with increased depth and breadth. In the realm of numerical ocean forecast modeling, the development of pre- and post-processing techniques can help mitigate the inherent biases found in ocean models. We expect that a hybrid approach, combining data-driven and dynamical numerical models will play a pivotal role in pushing the boundaries of excellence in ocean prediction.

*Code and data availability.* The codes used in this study are openly available at https://github.com/magray-ncsu/OceanNet. Data used in this study is available upon request.

*Author contributions.* All authors contributed equally to the research conducted and the writing of the paper.

*Competing interests.* None of the authors declare any competing interests.

*Acknowledgements.* We thank Subhashis Hazarika and Maria Molina for the insightful discussions, Jennifer Warrillow for editorial assis-
tance, and Elisabeth Brown and Gary Lackmann for their rigorous critiques of this manuscript. M. Gray, A. Lowe, T. Wu, and R. He, were supported by NSF awards 2019758 and 2331908. A. Chattopadhyay performed partial work at the Palo Alto Research Center, SRI International.



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
