# Peer review of "Long-term Prediction of the Gulf Stream Meander Using OceanNet: a Principled Neural Operator-based Digital Twin"

_EGUsphere, 2024_

## Referee Comment (RC1)

Review of the paper:

*Long-term Prediction of the Gulf Stream Meander Using OceanNet:*
*a Principled Neural Operator-based Digital Twin*
*by M. Gray et al.*

**1. General comments**

The paper introduces a new deep-learning based model, **OceanNet**, to predict Sea Surface Height (SSH) fields with physically reasonable predictions over at least 60 days. The authors used a high-resolution ocean reanalysis dataset (1993-2022) to train different neural architectures of their model over a Gulf Stream related domain. They also use Regional Ocean Model Simulations (ROMS) as a baseline for benchmarking in their tests. The paper is overall well written and presents promising results to improve the SSH predictability over a Gulf Stream related area., based on state-of-the-art deep learning strategies. I recommend publication after minor review taking into account some additional questions (2) and correction of typing errors (3). More precisely, I think the authors should consider to improve the presentation of Sections 2.2 to 2.4 by clearly stating from the beginning of these Sections, within a small paragraph for instance, which architectures are tested amongst all the developments. Giving clear names to each architecture would also be helpful I think. An additional table to summarize all their developments (UNet, UNet/integration scheme, UNet/integration scheme with shorter timesteps, FNO) would also help. The same applies for the Section Results where only Figures are displayed but a summary Table would be valuable.

**2. Specific comments:**

**Q.1: l.65.** Even if this is more explained in the following sections, can you precise from here what you intend by "predictability" of the SSH, which can be a very general formulation?

**Q.2: l.96.** In the paper, you state that you used EnKF for producing the reanalysis, which has no knowledge of future observations. I am not sure that it would be a problem to have reanalysis with knowledge from both past and future observations. In the end, what you need for training OceanNet is a sequence of Ground Truth (GT) that you will use to compare OceanNet predictions vs GT. So having the best reanalysis available would be compatible with your framework no?

**Q.3: l.96.** I would clearly state this as an Equation in the text, something like:
$x^*_{t+4} = DLOP(x^*_t)$

**Q4.** In the legend of Fig.3, the training losses are not yet introduced. You may want to let the readers know that all the related notations are defined in Section 2.5.

**Q5. l.172.** You precise that the model has 80*10e6 trainable parameters which is rather big. Can you give some information here or in Section 2.5 about the current computational capabilities you needed to train the model and how long it is (not only for inference which is very fast)?

**Q6. Fig.4**. Can you provide some complementary analyses regarding the plateau reached at day 40 by residual geostrophy? You mentioned it for PV but not for residual geostrophy. I guess there is a clear explanation why the plateau is reached by geostrophy at the same day that the model stops agreeing with reanalysis PV  (since they are connected). Maybe for the reader not a specialist in oceanography, it would be great to precise how these variables are connected.

**Q7. l199.** After reading the entire Section 2, I got the feeling that it would be valuable to add a Table here to summarize all the 4 architectures (*2 with the loss functions after but not yet presented at this point of the paper) tested among DLOP and OceanNet with specific architectures, pros and cons. This would also help the reader having an overview of the methods tested here.

**Q8. Eqs 5a and Eqs 5b.** You did not precise what are the notations $\hat{F}x$ and $\hat{F}y$, even if we easily understand this corresponds to the Fourier decomposition. The same for $|\hat{F}x|_k$ for the kth mode.

**Q9. Eq.6.** It is not clear to me how $a$ and $b$ are defined. You state that they are defined to agree with the MSE order of magnitude, which is clear but is it defined once and for all during training while the MSE magnitude may vary from batch to batch no? It is always tricky to define some weighting between multiple losses for sure but some comments added on that point would be great. What would happen if MSE prevails on spectral regularization? And the opposite? Would these parameters could have been trainable?

**Q10. l.290.** Similarly to Q.7, adding a Table to intercompare the 8 models, together with ROMS and persistence, with the metrics presented would be a good way to summarize everything, not only with Figs.7-9.

**Q11. Conclusion.** You identified the generalization of your work as an issue. Can you provide some ideas to overcome this problem? Maybe a decomposition of the global domain with specific training for each area? Even better, do you think there is a way to inject more information as inputs. For the specific SSH prediction, what would be great to add for instance to ease transfer learning on other domains (atmospheric forcings?, addition of physical constraints in the losses?)

**3. Technical corrections**

Please find below a list of grammatical or typing errors to consider before publication:

**l.27. reattached the to  GSM** -> to the
**l.175. What is the arrow in front of Eq.3?**
**l.251. an example of it can... ->** One example can

**l.254. validations images** -> may be replaced by ground truth reanalyses?

**l.276. that that**

**l.278. If the means of two objects of comparisons…** -> replace by: if the average metric of two models…?

**l.282. can still outperforms** -> outperform

---

## Referee Comment (RC2)

General Comments:

This paper addresses a very relevant topic, within the scope of Ocean Science, and does a very interesting job of assessing this. I do question if the work might be better placed in Geoscientific Model Development, but it certainly isn't out of place here.
The novel concept of data-driven ocean models is discussed, with a focus on the Gulf Stream through use of (predominantly) a Fourier neural operator based model, and with some comparison to a UNet based approach.
Methods are solid, a thorough assessment of results is given, and suitable conclusions derived. However, I feel results need to be caveated more given the limitations of the ROMS comparison used (see specific comments below).
Overall the work is really interesting. The paper is nicely presented, on the whole the narrative is clear, and figures compliment the text.
I feel the paper would benefit from a little more rigour in places. Specifically, labels and text on figures need a bit of careful consideration (see some specific notes below). Most importantly, the mathematics needs to be reassessed to ensure it is precise. It's important that throughout the paper variables are clearly explained, and used consistently, especially with differentiation between predicted values and true values (it seems that in places the same descriptor is used interchangeably for the true value and predicted value --- use of the hat symbol would greatly help readability in that respect).

Specific Comments:

Currently the abstract lacks information on performance and impact. For example, the computational gains of OceanNet are significant but not mentioned here, neither are the results.

Line 8, As far as I can tell the model doesn't track, or identify, the GSM, but predicts it. This is key and should be the focus, i.e. 'trained to predict...' rather than 'trained to identify and track...'.

Figure 1: It's not clear why all the subpanels cover different dates and/or averaging periods. It would be more cohesive if they were all the same (possibly separating into 2 figures, one with long term mean and one with an example daily mean), unless there's a sensible reason for the different views, in which case briefly explaining this would be useful.

Line 68-70 claims there are no global data-driven ocean models. This instead should cite Wang 24 and the XiHe model, https://arxiv.org/abs/2402.02995 which is a global data-driven ocean model.
More generally there are a growing number of related ocean approaches, it would be beneficial to include some of these, i.e. Xiong23, Bire23, Subel24.

Line 99: What is meant by 'as would be the case for observations'? Please can the authors expand a bit on this. Do they mean as would be the case in a realistic forecast scenario, rather than when training on past data?

Line 109-115: Please add a sentence making it clear what inputs and outputs are to the model --- it seems this is just a 2d SSH field for input (no other variables, and no temporal info, or explicit spatial info etc), and SSH field output. Clarifying this would be helpful.

Figure 2: It seems from the caption that 'Run 57' refers to the detail of the DLOP run, and has no meaningful relevance to the reanalysis. Ideally this would be replaced with the date which is being assessed, but otherwise omitted or used in the DLOP title, but not the reanalysis title.

Line 155-156: I think this misses the very important step of the convolution being applied? i.e. a "Fourier transform is performed... *convolutions are applied in Fourier space*... an inverse Fourier transform...."

Equation 2: I want to clarify if N and H as referred to in figure 3 and elsewhere (i.e. line 171) are *exactly* the parts of the equation as described in equation 2, or if they (at least N, possibly H as well) are the learnt approximations to this. It seems that we learn N (and to some extent maybe H), and so some differentiation between their true meaning/value, and the learnt approximation of them is needed if so. (Otherwise, if the learnt value as described in figure 3, and the use as described in equation 2 are *exactly* the same thing, where in the set up is the approximation, and where do errors come in when calculating X(t+delta_t)?)
Similarly, equation 3 should clarify that this is the approximation to X(t+delta_t), for example with use of a ^.

Figure 3 is quite busy. I think it would benefit from the loss function being included as an equation rather than within the figure.

Line 176-180: Again please clarify what the model inputs are, and what the model outputs are, and if PV and GEO are calculated from output variables, or explicitly output by the model (I assume the former, but it would be good to be explicit about this).

Line 176-187 and figure 4: this comparison could do with a baseline or comparison to clarify what 'good' means here. Can an example from an alternative method be added? (perhaps not, given persistence is not of any interest here, and numerical models are built to conserve so an unfair comparator. But some way of clarifying whether these differences are large or small in comparison to some scale of impact, or of current predictability, would be useful if one can be found).
Also, the framing of the discussion on conservation (Lines 176-187 and fig. 4) is confusing to me. Is this related to the DLOP, or the FNO model. Or is this something between the two? If the FNO model, then there is no need to refer to the DLOP model. If its related to the DLOP model and isn't the FNO, then it would seem more sensible to discuss this earlier in the paper. Clarification as to whether this model is an expansion of the FNO, or an expansion of the DLOP, and the exact nature of the difference would be helpful please.

Equation 4a and 4b: Use of N vs N1 is unclear to me. What's the difference here? Particularly re the left hand side of equation 4b.

It also seems that H is a function of X(t) as well as of N(X(t)). I think this needs a bit more thought, and mathematical rigour, both here and throughout the paper (i.e. in the description of the loss function, in figure 3, etc), ensuring it is consistent, clear, and correct throughout.

Is it best to define that H is applied to N? Or better to say that H is a function applied to X, and that N is a part of that function, i.e.

$$X_{t+1} = H(X_t) = X_t + \mathcal{N}\left(X_t + \frac{1}{2}\mathcal{N}(X_t, \varphi), \varphi\right)$$

With $\mathcal{N}(X_t, \varphi)$ is the output from…

Line 201 describes that the training data undergoes a 5 day mean processing step. Is this meaned data what is used as 'truth' for comparison, and in the loss function etc, or is the time-meaned data only used as training inputs? (I would hope the former). Can this be clarified here please, i.e. something like 'all data used for training, validation and testing undergoes a 5 day running mean…'

Line 260-262: It's stated that ROMs would be run with persistence if no atmospheric forecast was available, I've never heard of this being the case, especially for 120 day forecast periods. Ocean models are commonly forced using predictions from atmospheric models, and regional models are forced at the boundary by global models. I feel it should be noted in the text that this is not a realistic comparator to common use cases for this kind of prediction problem. This set up of ROMS may serve as an additional baseline comparator (alongside climatology and persistence), but the limitations of this comparison should be much better noted (or ideally, forecasts from the coupled version of ROMS, run with boundary forcing from a wider domain model, should be used for the full forecast period). There's a big difference between no forcing, as applied to the ML methods, and incorrect forcing (which is the case if using persistence for 120 days to force ROMS).

I don't think ROMS would be used in this way for predicting over 120 days (or even over 10 days), but if there are examples of this it would be good to reference them in the text here.

Figure 8: I think the pdf on the left is unnecessary and therefore a bit confusing. If kept it needs more clarification as to what it is (presumably, it's what was used to calculate the saturation metric?), but I don't think it's needed.

Figure 9, 10, 11 need a bit more clarity – having the legend on only the FNO plot is misleading, better to add this as a label down the side, similar to the DLOP/FNO labels across the top. Or at least as a legend on both DLOP and FNO plots.

In figure 9, why don't panes c and d have the value at day 60? Presumably because of the level of instability, but worth noting this briefly in the caption, assuming it's related to the instability of this model.

Figure 9 caption (and various places through the paper, i.e. figure 12) refer to the various runs as ensemble members, instead I would recommend using 'model runs' or

something else. Ensemble members predict for the same period and are in some way perturbed to give a variety of predictions for one specific time, but this isn't what's being done here as far as I can tell. These instead are multiple instances of model runs, from different start times. Using 'ensemble members' makes it confusing as to what they are.

Line 288: there isn't a run with *no* integration scheme ('a lack thereof'), presumably a simple euler first order is used, or some simple addition? The most basic form of integration is still an integration scheme. It needs to be clarified here what that was, even if its trivial.

Line 292 -294 Needs a bit of clarification.

Line 331 needs to be clear that this is ROMS with incorrect persistence forcing, rather than the more meaningful application of ROMS.

Line 342-345: This sentence doesn't make sense to me. More importantly, again, regional models are most often forced with predictions from global models. I don't know of any cases where predictions (especially long sub-seasonal to seasonal), are made using persistence for boundary conditions, as seems to be implied here.
My bigger concern though is the persistence for atmospheric forcing --- I suspect this has even greater impact, especially on SSH fields, than the boundary forcing, and isn't mentioned at all here. I think the caveat needs to be clear, and include the atmospheric aspect.

Technical comments:
In many places (i.e. Lines 3, 55, 115, 117, 132, 139, 252, and very possibly elsewhere) a hyphen is used when it should be an em-dash. This needs to be checked throughout the paper, not just the lines mentioned here as my search was not extensive.

Spelling of Chattopadhyay for 2023 paper missing the first a throughout; in both the bibliography and when cited in text (i.e. it seems an issue in the way its stored in the referencing software)

Line 86/87 citation for He, and Wu and He, are both missing years.

Line 86, I don't think the reference to figure 1 is relevant here, without a mention to the domain of OceanNet.

Line 127, implies lower CC, I assume this isn't the case(!)

Line 129-131, '....an inaccuracy of interest...'. This sentence is very hard to process, can it be amended please.

Equation 5a and 5b need $\hat{\mathcal{F}}$ to be defined.

Line 242, I would say these metrics are '...widely employed in assessment of data driven atmospheric forecasts' (rather than just 'forecasting'), given the references being pointed to.

Line 250, simply say 'an example of this contour can be seen in figure 1b'. The comment about it not being shown in other figures in the paper is unnecessary and a bit confusing.

Line 270, clarify that the data is (presumably) spatially averaged before averaging across multiple runs.

Line 282 outperforms -> outperform

Line 259, comma needed: '...compared were MSE, and MSE with...'

---

## Author Comment (AC1)

Response to Anonymous Reviewer #1

*We thank reviewer 1 for your thorough and thoughtful review. We greatly appreciate your positive feedback on our work and your helpful suggestions for improving the clarity and organization of our manuscript. Our point-by-point responses to your comments are detailed below.*

Specific comments:

- Q.1: l.65. Even if this is more explained in the following sections, can you precise from here what you intend by "predictability" of the SSH, which can be a very general formulation?

*To improve the clarity, we have revised this sentence as follow:*

*"This method achieved valuable results, showing improved prediction accuracy of SSH fields in the Gulf of Mexico for up to 12 weeks compared to persistence, which was used as the baseline".*

- Q.2: l.96. In the paper, you state that you used EnKF for producing the reanalysis, which has no knowledge of future observations. I am not sure that it would be a problem to have reanalysis with knowledge from both past and future observations. In the end, what you need for training OceanNet is a sequence of Ground Truth (GT) that you will use to compare OceanNet predictions vs GT. So having the best reanalysis available would be compatible with your framework no?

*We acknowledge that these sentences are confusing. We have revised them as follows:*

*"Unlike the 4D-var method, the EnKFDA method does not rely on future timestep observations or require forward and adjoint model iterations during data assimilation. This approach enables the efficient creation of a data-assimilative ocean reanalysis, allowing OceanNet to be trained on a time-space continuous reanalysis dataset."*

- Q.3: l.96. I would clearly state this as an Equation in the text, something like: x*_{t+4} = DLOP(x*_t)

*For the sake of clarity, we have updated this sentence as suggested:*

*"If X(t) is the initial five-day mean field of SSH at timestep t, then X(t + Δt) = DLOP(X(t)),*

*where Δt was determined during training to be four days."*

- Q4. In the legend of Fig.3, the training losses are not yet introduced. You may want to let the readers know that all the related notations are defined in Section 2.5.

*We have revised the caption of Figure 3 to include a statement recommended by the reviewer:*

*"The loss function is discussed in Sect. 2.5.1"*

- Q5. l.172. You precise that the model has 80*10e6 trainable parameters which is rather big. Can you give some information here or in Section 2.5 about the current computational capabilities you needed to train the model and how long it is (not only for inference which is very fast)?

*Following the reviewer's suggestion, we have included the training time information, but have positioned it at the end of the introduction.*

*"OceanNet, on average, outperforms SSH predictions made by the state-of-the-art Regional Ocean Modeling System (ROMS) across a 120-day period while maintaining a computational cost that is 4,000,000x cheaper (ROMS:10 hours across 144 CPUs; OceanNet: 1.18 seconds on a single NVIDIA A100 GPU) following a training period of approximately 12 hours (on an NVIDIA A100 GPU with 40GB of memory).".*

- Q6. Fig.4. Can you provide some complementary analyses regarding the plateau reached at day 40 by residual geostrophy? You mentioned it for PV but not for residual geostrophy. I guess there is a clear explanation why the plateau is reached by geostrophy at the same day that the model stops agreeing with reanalysis PV (since they are connected). Maybe for the reader not a specialist in oceanography, it would be great to precise how these variables are connected.

*As also pointed out by Reviewer 2, the framing of Figure 4 and its discussion is not as clear, and more importantly, it is not the focus of this study. We have decided to removed Figure 4 and its related discussion in the revision.*

- Q7. l199. After reading the entire Section 2, I got the feeling that it would be valuable to add a Table here to summarize all the 4 architectures (*2 with the loss functions after but not yet presented at this point of the paper) tested among DLOP and OceanNet with specific architectures, pros and cons. This would also help the reader having an overview of the methods tested here.

*Following the reviewer's suggestion, we have include a table of architecture configuration in the revision.*

- Q8. Eqs 5a and Eqs 5b. You did not precise what are the notations \hat{F}x and \hat{F}y, even if we easily understand this corresponds to the Fourier decomposition. The same for |\hat{F}x|_k for the kth mode.

*Following the reviewer's suggestion, we have updated the notation in the revision*

- Q9. Eq.6. It is not clear to me how a and b are defined. You state that they are defined to agree with the MSE order of magnitude, which is clear but is it defined once and for all during training while the MSE magnitude may vary from batch to batch no? It is always tricky to define some weighting between multiple losses for sure but some comments added on that point would be great. What would happen if MSE prevails on spectral regularization? And the opposite? Would these parameters could have been trainable?

*We have clarified this in the revision:*

*"Coefficients a and b are scaling factors used to ensure the order of magnitude of $\mu_x$ agrees with the order of magnitude of $\mu_y$ as well as the magnitude of the MSE loss (Eq. 7). Both a and b were determined via hyperparameter optimization to be 0.25"*

- Q10. l.290. Similarly to Q.7, adding a Table to intercompare the 8 models, together with ROMS and persistence, with the metrics presented would be a good way to summarize everything, not only with Figs.7-9.

*Following the reviewer's suggestion, we have included a table in the revision.*

- Q11. Conclusion. You identified the generalization of your work as an issue. Can you provide some ideas to overcome this problem? Maybe a decomposition of the global domain with specific training for each area? Even better, do you think there is a way to inject more information as inputs. For the specific SSH prediction, what would be great to add for instance to ease transfer learning on other domains (atmospheric forcings?, addition of physical constraints in the losses?)

*An approach for OceanNet's generalization is to implement it for the global ocean. Our team has been working on this and will report the results in a sperate correspondence. We have included the following sentence in the conclusion as a next step:*

*"Efforts to apply OceanNet to the global ocean are currently underway by our research team and will be reported in a future correspondence".*

3. Technical corrections

Please find below a list of grammatical or typing errors to consider before publication:

l.27. reattached the to GSM -> to the

l.175. What is the arrow in front of Eq.3?

l.251. an example of it can… -> One example can

l.254. validations images -> may be replaced by ground truth reanalyses?

l.276. that that l.278. If the means of two objects of comparisons… -> replace by: if the average metric of two models…?

l.282. can still outperforms -> outperform

*We thank the reviewer for this list. All have been corrected and incorporated in our revision.*

---

## Author Comment (AC2)

Response to Dr. Rachel Furner:

*We thank Dr. Rachel Furner for your thoughtful and constructive review. We greatly appreciate your recognition of the relevance of our work and your positive feedback on our methods and presentation.*

*Your suggestions are very valuable, and we have carefully addressed them in our manuscript revision and our responses below. We are grateful for your time and effort, which have helped us improve this manuscript.*

Specific Comments:

- Currently the abstract lacks information on performance and impact. For example, the computational gains of OceanNet are significant but not mentioned here, neither are the results.

*Following reviewer's suggestion, we have added the following sentence to the abstract:*

*"OceanNet can generate a 120-day forecast of the Gulf Stream Meander within seconds, offering significant computational efficiency."*

- Line 8, As far as I can tell the model doesn't track, or identify, the GSM, but predicts it. This is key and should be the focus, i.e. 'trained to predict…' rather than 'trained to identify and track…'.

*Following the reviewer's suggestion, we have revised the sentence as follows:*

*"…OceanNet (a neural operator-based digital twin for regional oceans) was trained to predict the GS's frontal position over subseasonal-to-seasonal timescales."*

- Figure 1: It's not clear why all the subpanels cover different dates and/or averaging periods. It would be more cohesive if they were all the same (possibly separating into 2 figures, one with long term mean and one with an example daily mean), unless there's a sensible reason for the different views, in which case briefly explaining this would be useful.

*Different dates were selected to highlight the dominant ocean feature in each sub-region. For the Gulf of Mexico, the focus of OceanNet is to predict the Loop Current and the Loop Current Eddies, which are not visible in the mean fields of this region. The date for the Gulf Stream Meander image was chosen to display a continuous feature with eddies on both sides of the northern boundary.*

- Line 68-70 claims there are no global data-driven ocean models. This instead should cite Wang 24 and the XiHe model, https://arxiv.org/abs/2402.02995 which is a global data-driven ocean model. More generally there are a growing number of related ocean

approaches, it would be beneficial to include some of these, i.e. Xiong23, Bire23, Subel24.

*Following the reviewer's comment, we have revised this sentence to include the reference of Xihe.*

*"These studies are certainly a step in the right direction, but there is ample need for more data-driven ocean models (e.g., Wang et al., 2023) similar to the data-driven global weather models seen in Pathak et al. (2022), Bi et al. (2023b), and Lam et al. (2023)"*

- Line 99: What is meant by 'as would be the case for observations'? Please can the authors expand a bit on this. Do they mean as would be the case in a realistic forecast scenario, rather than when training on past data?

*For clarification, we have revised this sentence as follows in the revision:*

*"Unlike the 4D-var method, the EnKFDA method does not rely on future timestep observations or require forward and adjoint model iterations during data assimilation. This approach enables the efficient creation of a data-assimilative ocean reanalysis, allowing OceanNet to be trained on a time-space continuous reanalysis dataset."*

- Line 109-115: Please add a sentence making it clear what inputs and outputs are to the model --- it seems this is just a 2d SSH field for input (no other variables, and no temporal info, or explicit spatial info etc), and SSH field output. Clarifying this would be helpful.

*Following the reviewer's suggestion, we have revised this sentence as follows in the revision:*

*"DLOP is a relatively simple U-Net and overall simpler than DLWP, but the core idea is the same: pixel-wise connections of two-dimensional fields of physical variables (i.e., SSH fields in this study) between timesteps are sufficient to predict the evolution of such fields (SSH in this study) through time."*

- Figure 2: It seems from the caption that 'Run 57' refers to the detail of the DLOP run, and has no meaningful relevance to the reanalysis. Ideally this would be replaced with the date which is being assessed, but otherwise omitted or used in the DLOP title, but not the reanalysis title.

*Following the reviewer's suggestion, we have changed the title of the subplots to reflect the date they are valid*

- Line 155-156: I think this misses the very important step of the convolution being applied? i.e. a "Fourier transform is performed… *convolutions are applied in Fourier space*… an inverse Fourier transform…."

*In our design of FNO, convolutions are not applied in the Fourier space. Please see Figure 3.*

- Equation 2: I want to clarify if N and H as referred to in figure 3 and elsewhere (i.e. line 171) are *exactly* the parts of the equation as described in equation 2, or if they (at least N, possibly H as well) are the learnt approximations to this. It seems that we learn N (and to some extent maybe H), and so some differentiation between their true meaning/value, and the learnt approximation of them is needed if so. (Otherwise, if the learnt value as described in figure 3, and the use as described in equation 2 are *exactly* the same thing, where in the set up is the approximation, and where do errors come in when calculating X(t+delta_t)?) Similarly, equation 3 should clarify that this is the approximation to X(t+delta_t), for example with use of a ^.

*We acknowledge that some of our choices of notation for equations and figures were inconsistent. Per the reviewer's suggestion, we have adopted notation that is more common in machine learning disciplines. We have reviewed all equations and figures for consistency.*

- Figure 3 is quite busy. I think it would benefit from the loss function being included as an equation rather than within the figure.

*We acknowledge that Figure 3 is quite detailed, but we think it effectively consolidates all the relevant information in one place for the convenience of readers. After careful consideration, we have decided to retain the current format.*

- Line 176-180: Again please clarify what the model inputs are, and what the model outputs are, and if PV and GEO are calculated from output variables, or explicitly output by the model (I assume the former, but it would be good to be explicit about this).

*We have removed this section in the revision. Please see our response to the next comment of the reviewer.*

- Line 176-187 and figure 4: this comparison could do with a baseline or comparison to clarify what 'good' means here. Can an example from an alternative method be added? (perhaps not, given persistence is not of any interest here, and numerical models are built to conserve so an unfair comparator. But some way of clarifying whether these differences are large or small in comparison to some scale of impact, or of current predictability, would be useful if one can be found). Also, the framing of the discussion on conservation (Lines 176-187 and fig. 4) is confusing to me. Is this related to the DLOP, or the FNO model. Or is this something between the two? If the FNO model, then there is no need to refer to the DLOP model. If its related to the DLOP model and isn't the FNO, then it would seem more sensible to discuss this earlier in the paper. Clarification as to whether this model is an expansion of the FNO, or an expansion of the DLOP, and the exact nature of the difference would be helpful please.

*We agree with the reviewer that the framing of the discussion on conservation (Lines 176-187 and fig. 4) is not as clear, and more importantly, it is not the focus of this study. We have removed Figure 4 and its related discussion in the revision.*

- Equation 4a and 4b: Use of N vs N1 is unclear to me. What's the difference here? Particularly re the left hand side of equation 4b. It also seems that H is a function of X(t) as well as of N(X(t)). I think this needs a bit more thought, and mathematical rigour, both here and throughout the paper (i.e. in the description of the loss function, in figure 3, etc), ensuring it is consistent, clear, and correct throughout. Is it best to define that H is applied to N? Or better to say that H is a function applied to X, and that N is a part of that function, i.e. $X_{t+1} = H(X_t) = X_t + \mathcal{N}(X_t + \frac{1}{2}\mathcal{N}(X_t, \varphi), \varphi)$ With $\mathcal{N}(X_t, \varphi)$ is the output from…

*We have clarified this by adding to the revision, ""…where N1 represents the output of the model from the previous timestep"*

- Line 201 describes that the training data undergoes a 5 day mean processing step. Is this meaned data what is used as 'truth' for comparison, and in the loss function etc, or is the time-meaned data only used as training inputs? (I would hope the former). Can this be clarified here please, i.e. something like 'all data used for training, validation and testing undergoes a 5 day running mean…'

*The reviewer is correct about the 5-day running means. Following your suggestion, we have revised this sentence to:*

*"OceanNet for the GSM was trained on five-day running mean SSH reanalysis fields from 1993 to 2018, which helped remove high-frequency features like tides. The years 2019 and 2020 were reserved for validation and testing. All data used for training, validation, and testing underwent the same five-day running mean procedure."*

- Line 260-262: It's stated that ROMs would be run with persistence if no atmospheric forecast was available, I've never heard of this being the case, especially for 120 day forecast periods. Ocean models are commonly forced using predictions from atmospheric models, and regional models are forced at the boundary by global models. feel it should be noted in the text that this is not a realistic comparator to common use cases for this kind of prediction problem. This set up of ROMS may serve as an additional baseline comparator (alongside climatology and persistence), but the limitations of this comparison should be much better noted (or ideally, forecasts from the coupled version of ROMS, run with boundary forcing from a wider domain model, should be used for the full forecast period). There's a big difference between no forcing, as applied to the ML methods, and incorrect forcing (which is the case if using persistence for 120 days to force ROMS). I don't think ROMS would be used in this way for predicting over 120 days (or even over 10 days), but if there are examples of this it would be good to reference them in the text here.

*In regional ocean forecasting, defining surface and boundary forcing is a significant challenge, particularly when accurate and continuous global ocean and atmosphere forecasting data for extended periods is unavailable. Currently, almost all global ocean and atmosphere forecasts extend only 7-10 days, whereas our research requires forecasts with a*

*duration of 120 days. None of the operational global models routinely provide such long-term forecasts.*

*Persistence, in this context, refers to the assumption that future conditions will resemble past conditions. This approach can be used to define boundary and surface forcing for regional-scale models. However, as the reviewer correctly pointed out, it is important to understand the limitations of persistence and how it is applied in practice.*

*Persistence is commonly used in short- to medium-term ocean forecasting due to its simplicity (e.g., Jacox et al., 2020). However, it does not account for changes in climatic conditions, such as those driven by El Niño or other large-scale climate phenomena. While persistence can provide a baseline, it is not expected to capture full variability or trends in long-term forecasts.*

*We acknowledge the limitation of using persistent forcing in driving ROMS forecasts. This limitation lies not with ROMS, as a dynamical model, but with the specific ROMS forecast configuration that we adopted in this study. We have incorporated this discussion and the reference below in the revision, and we thank the reviewer for this insightful comment.*

Reference:

Michael G. Jacox, Michael A. Alexander, Samantha Siedlecki, Ke Chen, Young-Oh Kwon, Stephanie Brodie, Ivonne Ortiz, Desiree Tommasi, Matthew J. Widlansky, Daniel Barrie, Antonietta Capotondi, Wei Cheng, Emanuele Di Lorenzo, Christopher Edwards, Jerome Fiechter, Paula Fratantoni, Elliott L. Hazen, Albert J. Hermann, Arun Kumar, Arthur J. Miller, Douglas Pirhalla, Mercedes Pozo Buil, Sulagna Ray, Scott C. Sheridan, Aneesh Subramanian, Philip Thompson, Lesley Thorne, Hariharasubramanian Annamalai, Kerim Aydin, Steven J. Bograd, Roger B. Griffis, Kelly Kearney, Hyemi Kim, Annarita Mariotti, Mark Merrifield, Ryan Rykaczewski (2020), Seasonal-to-interannual prediction of North American coastal marine ecosystems: Forecast methods, mechanisms of predictability, and priority developments, Progress in Oceanography, Volume 183,2020, 102307, ISSN 0079-6611, https://doi.org/10.1016/j.pocean.2020.102307.

- Figure 8: I think the pdf on the left is unnecessary and therefore a bit confusing. If kept it needs more clarification as to what it is (presumably, it's what was used to calculate the saturation metric?), but I don't think it's needed.

*Yes, the PDFs on the left side of Figure 8 are used to calculate the saturation metrics, which determined 95% of the means derived from the random pair. We have followed the reviewer's suggestion and removed the PDFs on the figure. The figure (now figure 7) retains the saturation value on the plot, but is explained more thoroughly in the caption:*

*"OceanNet's performance metrics in the northwest Atlantic: RMSE (top), CC (middle), and MHD (bottom), compared to the persistence forecast and ROMS dynamical model forecast. The performance statistics, calculated based on forecasts of 0-120 days, are displayed as mean values (lines) with standard deviations (shading). The black horizontal dashed lines denote saturation values, which are determined as 95% of the means derived from 1,000 pairs of*

*random images in the reanalysis dataset. These representations illustrate how each method's statistics compare with the target SSH from the reanalysis dataset".*

- Figure 9, 10, 11 need a bit more clarity – having the legend on only the FNO plot is misleading, better to add this as a label down the side, similar to the DLOP/FNO labels across the top. Or at least as a legend on both DLOP and FNO plots.

*Following the reviewer's suggestion, we have revised Figure 9-11 by adding legends to DLOP plots as well.*

- In figure 9, why don't panes c and d have the value at day 60? Presumably because of the level of instability, but worth noting this briefly in the caption, assuming it's related to the instability of this model.

*Due to instabilities in the model runs shown in (c) and (d), their RMSE values are so large that they cannot be meaningfully represented in the plot. We have noted this issue in the revised caption of Figure 9.*

- Figure 9 caption (and various places through the paper, i.e. figure 12) refer to the various runs as ensemble members, instead I would recommend using 'model runs' or something else. Ensemble members predict for the same period and are in some way perturbed to give a variety of predictions for one specific time, but this isn't what's being done here as far as I can tell. These instead are multiple instances of model runs, from different start times. Using 'ensemble members' makes it confusing as to what they are.

*Following the reviewer's suggestion, we have changed "ensemble members" to "model runs" in the revision.*

- Line 288: there isn't a run with no integration scheme ('a lack thereof'), presumably a simple euler first order is used, or some simple addition? The most basic form of integration is still an integration scheme. It needs to be clarified here what that was, even if its trivial.

*We have refined this sentence in the revision to improve the readability:*

*"The RMSE, anomaly correlation coefficient (ACC), and MHD are compared across different iterations of the DLOP and FNO models, focusing on integration schemes and loss function terms. The two integration schemes compared were the absence of integration and PEC. The loss function terms compared were MSE and MSE with spectral regularization. This combination of model types, integration schemes, and loss function terms results in eight models to compare, following the same approach as before (ensembled metrics, Fig. 7), against each other and with ROMS and persistence predictions."*

- Line 292 -294 Needs a bit of clarification.

*We have revised this sentence as follows, to improve its clarity:*

*"RMSE not only indicates the magnitude of values present but also serves as a measure of accuracy and stability. A high RMSE suggests that the magnitudes in the analyzed field are, on average, less realistic. If RMSE continues to increase over time, it implies that the model is becoming unstable."*

- Line 331 needs to be clear that this is ROMS with incorrect persistence forcing, rather than the more meaningful application of ROMS.

*Please see our responses above to reviewer's comments on line 260-262.*

- Line 342-345: This sentence doesn't make sense to me. More importantly, again, regional models are most often forced with predictions from global models. I don't know of any cases where predictions (especially long sub-seasonal to seasonal), are made using persistence for boundary conditions, as seems to be implied here. My bigger concern though is the persistence for atmospheric forcing --- I suspect this has even greater impact, especially on SSH fields, than the boundary forcing, and isn't mentioned at all here. I think the caveat needs to be clear, and include the atmospheric aspect.

*Please see our responses above to reviewer's comments on line 260-262.*

---

## Referee Report (RR1)

**2nd Review of Long-term Prediction of the Gulf Stream Meander Using OceanNet: a Principled Neural Operator-based Digital Twin**

Many of the comments from my original review have been suitably addressed, thank you. However, I feel there are still some elements which need to be addressed. Apologies for the length, I've tried to be very thorough so it's clear where I feel there are still issues and what these are.

For me it's still very unclear how the model works, and the response to my questions around the wording in section 2.4.1, figure 3, and the use of $N$ and H, still leave me quite confused. I feel figure 3 and section 2.4.1 need to be substantially updated to clarify the model architecture.

The description in section 2.4.1 doesn't seem to match with figure 3, and this, along with equations 1 and 2, mean it's unclear from the paper how the model works. In particular I have the following questions around this section/figure/equations:

- The only info in section 2.4.1 about the way the network itself works seems to be the sentence 'a Fourier transform is performed on the input data, the highest Fourier modes are reduced to zero, then an inverse Fourier transform brings the data back to a real space where it is concatenated with the input image'. To me, this implies the data is moved to Fourier space, reduced, and re-projected back to real space. I.e. a reduced representation of the data (still at time t) is the output of the network. This leaves the question as to where any prediction is happening (to either predict the increment to X(t), or to predict X(t+1)). Figure 3 however appears far more complicated, and includes far more than just FFT and IFFT (i.e. the schematic of an MLP, the box labelled R, and the concatenation of items) It seems the write up misses a lot of steps, and to me would greatly benefit from a broader explanation of the steps in the network in section 2.4.1.
- Figure 3a schematic implies a multi-layer perceptron (MLP) is used (the orange dots, connected with black lines), prior to the fourier layer, after the fourier layer, and within the Fourier layer. But nothing in section 2.4.1 mentions this. What is this part of the schematic and what is it doing? What are the input nodes in this MLP, and what is output (i.e. the dimensions of the output, and what does it represent, if anything)?
- Figure 3b is defined as the 'Fourier Layer', but the sideways curly brace under $N$, moving from part a to part b implies that the grey box in part b covers the entire process shown in part a. This is really confusing to me. If schematic 3b is a subset of schematic 3a, please can you clarify which of 3a is covered. I suspect that what's intended is that all of 3a combines give $N$, if this is the case, the curly brace needs to clearly start and end either side of the schematic (currently it starts and ends in the middle of the MLP), and most importantly, the brace needs to be above $N$, pointing at $N$. It would also help to then have a clear space between figure 3a and figure 3b. I would also suggest having $N$(x(t) at the right hand side of fig. 3a, to make it clear what the output is here.
- It's not clear what is being predicted by $N$. Is this simply giving a reduced representation of X(t), is it predicting the increment to X(t), or is this predicting X(t+1) in a way which is then adapted by H? Please can this be clearly stated in the text, and in figure 3 ( i.e. $N$(x(t) = Δx(t) )
- The use of $v$ in figure 3b is confusing, as the only explanation given is '$v$(t) represents some 2d field'. I think more clarification is needed as to what $v$ is - is this the input fields, or inputs processed in some way (the schematic implies $v$ is the output of the MLP, is this correct?). Is each single 2d field processed independently as implied at present, with no awareness of the others? Given this case deals with single 2d SSH field inputs, the schematic feels very complex, implying combinations of inputs create $v$.
- What is R in figure 3b? This doesn't seem to be mentioned anywhere.
- Equation 2 feels a bit confusing in its set up. Again, careful consideration needs to be given to the maths and descriptions here. In particular, there's an integral applied to F(X(t)), but then H is referred to as the integrator and covers the whole of the right hand side, not just the integral. It's not clear to me where the boundaries of $N$ should sit here - the curly brace starts after the integral, but includes dt. I would have thought either the integral (both the integral sign and dt) are included in $N$, or just the inside (F(X(t)) is included in $N$ (not just dt as is shown here). The use of $N$ later, especially the

discussion of the PEC scheme in section 2.4.2 implies $N$ is a prediction of the increment, i.e. it *includes* the integral sign in equation 2. And then H deals with how you add that increment on, whereas the current schematic doesn't give this. If this is the case, I think calling H the integrator needs to be carefully described to distinguish the way H is integrating from the integral sign used, perhaps referring to the time-stepping when explaining H.

- If leaving the loss function in figure 3, then the caption should explain what $N_0$ is please.

I commented previously on the mathematical use of H and N, and the need for mathematical rigour - The paper still states that H is a function of $N$ only. However this isn't the case, as X is a direct input to H (the first term on the right hand side in eq 4b).
I think this needs updating throughout the paper to replace H( $N(X(t),\Theta)$ ) with H( X(t), $N(X(t),\Theta)$ ).

Figure 7 has an erroneous " at the end.

The paper still refers to 'no integration scheme', and 'the absence of integration'. I cannot understand how this could be the case. If there is no integration how is the value changing? The model is timestepping somehow, as the blue line in figure 8 varies with time. It may be a very simple integrator is used, but the model is being integrated over time. This still needs correcting.
This perhaps speaks to a wider confusion over the use of the term 'integrator' through the paper - if this is being used to refer to the way the model moves from time step t, to time step t+1, given some output from the network, then perhaps this could be clarified when first used, to distinguish this from mathematical integration as used in the equations.
If, as I think is the case, 'integrator' is referring to how the model moves from one time step to the next, then there must be something used to do this in all versions of the model runs - any reference to 'no integrator' or similar needs to be updated throughout the paper, including in figure captions and tables etc. If I've misunderstood, then there needs to be some clarification in the paper as to what the 'integrator' means, as well as a description of how different versions of the model are being time stepped.

Line 350-355 currently reads: *"The comparisons between OceanNet and ROMS can also be considered to have a major caveat: ROMS as a regional ocean model depends on **persistent forcing conditions** at open boundaries, for which persistent boundaries were provided in this study. **While this method of modelling is operational in the sense** that the absence of subseasonal-to-seasonal prediction often lacks forcing information on similar timescales thus persistence must be used, it may be more fair to compare the performance of OceanNet to a model which does not require boundary conditions, such as a global ocean model."*
The first use of persistence here is incorrect - ROMS does not depend on *persistent* forcing. It depends on boundary and atmospheric forcing, which can be provided from a variety of sources, not simply from persistence (this forcing rarely comes from persistence, most commonly it comes from large models, or from climatological forecasts)
Secondly, as I mentioned in my first review, I feel it is very misleading to refer to the long term use of persistence forcing as 'operational' in any sense here. Having worked in operational regional oceanography I don't think ROMS would ever be used in this way in an operational setting, certainly I have never heard of anything like this. In almost all operational settings, a forecast from a broader model would be available and the model would be forced with this. I don't feel the comparison needs to be re-done, but this caveat needs to properly clarify that it is not being compared to anything resembling a regularly used configuration, and certainly not to anything operational. 'Operational' has a specific meaning in this context and does not seem to be correctly used here - the comments made in the paper imply a very poor approach is taken in operational ocean forecasting, which is not the case. Perhaps there is some confusion over the terminology here. In operational cases (and I would imagine almost all research cases) ROMS would not be used over these timescales with persistence forcing, far better options exist and would be used.
I would like to see this updated, with any reference to operational applications using persistence removed. I suggest something along the lines of the following:

'*The comparisons between OceanNet and ROMS can also be considered to have a major caveat: ROMS as a regional ocean model depends on **providing** forcing conditions at open boundaries, for which persistent boundaries were provided in this study. **This method would not be used in meaningful prediction scenarios over the timescales considered here, and as such** it may be more fair to compare the performance of OceanNet to a model that does not require boundary conditions, such as a global ocean model, **or to a configuration of ROMS forced with boundary conditions taken from predictions produced by a global ocean model.**"*

The authors also do not seem to have made any reference in these caveats to the use of persistence forcing for the atmospheric forcing conditions as I flagged as a concern in my first review. Again this element of the set up severely degrades the quality of the predictions from ROMS, and again ROMS would not be used to provide forecasts over these timescales with persistent atmospheric forcing. I think this caveat must be noted in the paper.

Following my review, the authors updated the paper to state: *"Currently, almost all global ocean and atmosphere forecasts extend only 7-10 days".* This is not correct, many operational forecast centres run forecasts for almost all lead times. Short term systems give forecasts for 7-10 days, but many forecast centres run operational sub-seasonal to seasonal (S2S) forecast systems (generally providing ocean and atmospheric forecasts within a coupled system) which predict out to a few months, and many other systems provide even longer forecasts, all the way out to mutli-decadal climate forecasts and beyond. The paper they cite (Jacox et al., 2020) seems to be a *research* application of marine biology predictions, not an operational system, and not forecasting the ocean itself. The incorrect statements over *operational* forecasts must be amended.

Similarly, the paper now states *"in practice. Persistence is commonly used in short- to medium-term ocean forecasting due to its simplicity (e.g., Jacox et (2020)), but it does not account for changes in climatic conditions such as those driven by El Niño or other large-scale climate."* Persistence is not 'commonly' used, and the issues with persistence are far greater than this - persistence forecasts do not account for *any* variability, over any timescales. For applications such as those in the paper it's not just the lack of climatic variability, but any weather impacts, such as winds, storms/hurricanes, warming or cooling over a few days/weeks etc. The caveat here does not accurately capture the significant limitations of the use of persistence.

In general, I think that paper has some really interesting science in it, but still needs notable work in describing the methods, and in being clear about the caveats of the comparison given. Along with clarification over the alternative methods to iterate/time-step the models used.

---

## Author Response (AR2)

**2nd Review of Long-term Prediction of the Gulf Stream Meander Using OceanNet: a Principled Neural Operator-based Digital Twin**

**NOTE: We have noticed an error in the numbering of sections. This has been updated. Please refer to the newest iteration of the manuscript for any mention of lines and sections by the authors.**

Many of the comments from my original review have been suitably addressed, thank you. However, I feel there are still some elements which need to be addressed. Apologies for the length, I've tried to be very thorough so it's clear where I feel there are still issues and what these are.

For me it's still very unclear how the model works, and the response to my questions around the wording in section 2.4.1, figure 3, and the use of $N$ and H, still leave me quite confused. I feel figure 3 and section 2.4.1 need to be substantially updated to clarify the model architecture.

The description in section 2.4.1 doesn't seem to match with figure 3, and this, along with equations 1 and 2, mean it's unclear from the paper how the model works. In particular I have the following questions around this section/figure/equations:

-   The only info in section 2.4.1 about the way the network itself works seems to be the sentence 'a Fourier transform is performed on the input data, the highest Fourier modes are reduced to zero, then an inverse Fourier transform brings the data back to a real space where it is concatenated with the input image'. To me, this implies the data is moved to Fourier space, reduced, and re-projected back to real space. I.e. a reduced representation of the data (still at time t) is the output of the network. This leaves the question as to where any prediction is happening (to either predict the increment to X(t), or to predict X(t+1)). Figure 3 however appears far more complicated, and includes far more than just FFT and IFFT (i.e. the schematic of an MLP, the box labelled R, and the concatenation of items) It seems the write up misses a lot of steps, and to me would greatly benefit from a broader explanation of the steps in the network in section 2.4.1.

We thank the reviewer for taking the time to dissect this section so thoroughly and provide us with areas of improvement.

-   The purpose of section 2.3.1 is to describe why the FNO is chosen and how it is employed theoretically for our problem. We leave the specifics of the FNO to the reference of Li et al. 2020 since we believe its complexity would distract the reader from the intent of the study. Thanks to the feedback from the reviewer, we have noticed the figure reference is incorrect on line 160. We added the following to section 2.3.1 to provide clarification on the FNO and have altered the original statement for continuity:
    -   Line 158-160:"Following the methods of Li et al. (2020), the Fourier layer takes a high-dimensional representation of the input field, applies a Fourier transform, reduces the highest Fourier modes to zero, and applies an inverse Fourier transform to bring the data back to its original space. The resulting tensor is then concatenated with the input to the Fourier layer, to which a 2D convolution has been applied to account for aperiodicity in the data (Fig.3b).
-   We are averse in changing the actual figure if it can be avoided since the figure is a reproduction of Figure 6 in our partner publication "OceanNet: A principled neural operator-based digital twin for regional oceans" (Chattopadhyay et al. 2023), which is conveyed to the reader (lines 81-84); however, we have updated the caption for Fig.3 to be clearer for the reader and better reflect the edits mentioned above:
    -   (a) A schematic of the OceanNet model with input image X(t). Prior to entering the Fourier layers, the input field is lifted to a higher dimensional space by means of two convolutional

layers. The data then flows through four Fourier layers. The output of each of the Fourier layers is activated with the Gaussian Error Linear Units function. Following the last Fourier layer, the data is fed through two more convolutions to preserve the dimensions of the final output. (b) The Fourier Neural Operator, depicted as N. A Fourier transform is performed on v(t), the higher-dimensional representation of the input image, followed by a linear operation, R, to reduce the highest Fourier modes to zero, resulting in v~(t). An inverse Fourier transform brings v~(t) back to its original space. The resulting tensor is then concatenated with the input to the Fourier layer, to which a 2D convolution has been applied.

- (c) and (d) are addressed below

- Figure 3a schematic implies a multi-layer perceptron (MLP) is used (the orange dots, connected with black lines), prior to the fourier layer, after the fourier layer, and within the Fourier layer. But nothing in section 2.4.1 mentions this. What is this part of the schematic and what is it doing? What are the input nodes in this MLP, and what is output (i.e. the dimensions of the output, and what does it represent, if anything)?

  o Thank you for catching this inconsistency between the figure and the text. Our proposed resolution for this comment can be found above.

- Figure 3b is defined as the 'Fourier Layer', but the sideways curly brace under $N$, moving from part a to part b implies that the grey box in part b covers the entire process shown in part a. This is really confusing to me. If schematic 3b is a subset of schematic 3a, please can you clarify which of 3a is covered. I suspect that what's intended is that all of 3a combines give $N$, if this is the case, the curly brace needs to clearly start and end either side of the schematic (currently it starts and ends in the middle of the MLP), and most importantly, the brace needs to be above $N$, pointing at $N$. It would also help to then have a clear space between figure 3a and figure 3b. I would also suggest having $N$(x(t) at the right hand side of fig. 3a, to make it clear what the output is here.

  o (thanks, etc). We are averse in changing the actual figure if it can be avoided since the figure is a reproduction of Figure 6 in our partner publication "OceanNet: A principled neural operator-based digital twin for regional oceans" (Chattopadhyay et al. 2024), which is conveyed to the reader (lines 81-84). We ask that this suggestion be disregarded so continuity between the two papers may be maintained for readers.

- It's not clear what is being predicted by $N$. Is this simply giving a reduced representation of X(t), is it predicting the increment to X(t), or is this predicting X(t+1) in a way which is then adapted by H? Please can this be clearly stated in the text, and in figure 3 ( i.e. $N$(x(t) = Δx(t) )

  o We appreciate the reviewer's attention to the intricacies of our model's theory and acknowledge that our original test was unclear in this regard. What is being predicted by N[X(t), \theta] is dependent on the integration scheme selected by the user. For example, if one elects to use the implicit Euler integration scheme, then the neural network, N, would be predicting the increment of X(t). In another example, if one was to choose no integration scheme and would instead train the neural network N to directly predict the next timestep, then N[X(t), \theta] would output X(t + \delta t). To clarify this in the paper, we have changed lines 181-182 to read the following: "In practical terms, a future timestep X(t + \delta t) is predicted by feeding the initial image X(t) into our neural network N with parameters \theta. The numerical integration scheme H is then applied to the outputs as discussed in Chattopadhyay & Hassanzadeh (2023)"

  o To further address this point, we have provided another example of an integration scheme in the section immediately following the above, section 2.3.2.

- The use of $\boldsymbol{v}$ in figure 3b is confusing, as the only explanation given is '$\boldsymbol{v}$(t) represents some 2d field'. I think more clarification is needed as to what $\boldsymbol{v}$ is - is this the input fields, or inputs processed in some way (the schematic implies $\boldsymbol{v}$ is the output of the MLP, is this correct?). Is each single 2d field processed independently as implied at present, with no awareness of the others? Given this case deals with single 2d SSH field inputs, the schematic feels very complex, implying combinations of inputs create $\boldsymbol{v}$.
    - Thank you for catching this inconsistency between the figure and the text. Our proposed resolution for this comment can be found above.
- What is R in figure 3b? This doesn't seem to be mentioned anywhere.
    - Thank you for catching this inconsistency between the figure and the text. Our proposed resolution for this comment can be found above.
- Equation 2 feels a bit confusing in its set up. Again, careful consideration needs to be given to the maths and descriptions here. In particular, there's an integral applied to F(X(t)), but then H is referred to as the integrator and covers the whole of the right hand side, not just the integral. It's not clear to me where the boundaries of $N$ should sit here - the curly brace starts after the integral, but includes dt. I would have thought either the integral (both the integral sign and dt) are included in $N$, or just the inside (F(X(t)) is included in $N$ (not just dt as is shown here). The use of $N$ later, especially the discussion of the PEC scheme in section 2.4.2 implies $N$ is a prediction of the increment, i.e. it *includes* the integral sign in equation 2. And then H deals with how you add that increment on, whereas the current schematic doesn't give this. If this is the case, I think calling H the integrator needs to be carefully described to distinguish the way H is integrating from the integral sign used, perhaps referring to the time-stepping when explaining H.

We appreciate the thoroughness of the reviewer in her following of the mathematics of our paper and have added clarifications to the text to improve the consistency of the terms used throughout:
    - We have carefully reviewed the mathematics throughout the submission and have found no computational errors. However, we do acknowledge that equation 2 has a typo that has been addressed: the bracket used for visually defining N[o,\theta] no longer includes dt.
    - We have only found one instance of H being referred to as an "integrator". In the caption of Figure 3c, we have updated the clause in question to read: "the two-time-step scheme with the numerical integration operator, H". H is originally defined in lines 178-179 as "some implicit integration scheme", which is not contradicted in the visual depiction in equation 2. Other references to H used are "integration scheme" and "operator", both of which we believe to be appropriate. To be more explicit in its definition, we have amended the introduction H to be the following: "H[o] represents an operator encompassing the numerical technique used to evaluate the right-hand side of Eq.2 and will henceforth be referred to as an the 'numerical integration scheme'."
    - We have found instances where the "PEC integration scheme" has been referred to as an "integrator." We have replaced such instances with "integration scheme" to remain consistent with the references to H
- If leaving the loss function in figure 3, then the caption should explain what $N_0$ is please.

    - Thank you for pointing out the lack of completeness in the caption of our figure. The caption of figure 3d has been updated to the following: "The point-wise loss function used, constructed by the spectral regularizer \mu and MSE L1 for M samples and applied to all N_o ocean points. The loss function is discussed in greater detail in section 2.4.1"

I commented previously on the mathematical use of H and N, and the need for mathematical rigour - The paper still states that H is a function of $N$ only. However this isn't the case, as X is a direct input to H (the first term on the right hand side in eq 4b).

I think this needs updating throughout the paper to replace H( $N$(X(t),$\theta$) ) with H( X(t), $N$(X(t),$\theta$) ).

- o (thanks, etc) We respectfully disagree with the reviewer's assessment of the uses of H and N throughout the article. H is explicitly referred to as an "operator" and an "integration scheme," not a function. While our selection of integration scheme is implicit and thus depends on both X(t) and N[X(t), \theta], this does not have to be the case in general. In section 3 we reference model configurations with "no integration scheme", which means the neural network is directly predicting the next timestep. In such a case, the output is only estimated by N[X(t), \theta]. The decision for our notation allows for consistency across all possible numerical integration schema. Please refer to our responses above for the improvements we have made to increase clarification throughout the text in this regard.

Figure 7 has an erroneous " at the end.

- o Thank you for correcting this error. The quotation mark in question has been removed.

The paper still refers to 'no integration scheme', and 'the absence of integration'. I cannot understand how this could be the case. If there is no integration how is the value changing? The model is timestepping somehow, as the blue line in figure 8 varies with time. It may be a very simple integrator is used, but the model is being integrated over time. This still needs correcting.

This perhaps speaks to a wider confusion over the use of the term 'integrator' through the paper - if this is being used to refer to the way the model moves from time step t, to time step t+1, given some output from the network, then perhaps this could be clarified when first used, to distinguish this from mathematical integration as used in the equations.

If, as I think is the case, 'integrator' is referring to how the model moves from one time step to the next, then there must be something used to do this in all versions of the model runs - any reference to 'no integrator' or similar needs to be updated throughout the paper, including in figure captions and tables etc. If I've misunderstood, then there needs to be some clarification in the paper as to what the 'integrator' means, as well as a description of how different versions of the model are being time stepped.

- o We thank the reviewer for her thorough investigation of our mathematics. We have addressed all instances of the use of the word "integrator" (see above). Our definition of N on lines 176-177 identifies N[o, \theta] as the neural network parameterizing F, regarding equation 2, while H is defined on line 178-179 as "an operator encompassing the numerical technique used to evaluate the right-hand side of Eq.2" (amended, see above response). H can be *any* numerical integration scheme (e.g. implicit Euler, Runge-Kuta 2nd order, Runge-Kuta 4th order, PEC, etc). To clarify the phrases "no integration scheme" and "the absence of integration", we have added the following after line 199: "As mentioned above, experiments were performed on a variety of models, some of which did not employ a numerical integration scheme in their methods. In such cases, the neural network N is directly predicting the next timestep, as is commonly seen in CNN and U-Net models such as those discussed in section 2.2.1. The equation representing such these models can be given as: X(t + \delta t) = H[N[X(t), \theta]] = N[X(t), \theta]"

Line 350-355 currently reads: *"The comparisons between OceanNet and ROMS can also be considered to have a major caveat: ROMS as a regional ocean model depends on **persistent forcing conditions** at open boundaries, for which persistent boundaries were provided in this study. **While this method of modelling is operational in the sense** that the absence of subseasonal-to-seasonal prediction often lacks forcing information on similar timescales thus persistence must be used, it may be more fair to compare the performance of OceanNet to a model which does not require boundary conditions, such as a global ocean model."*

The first use of persistence here is incorrect - ROMS does not depend on *persistent* forcing. It depends on boundary and atmospheric forcing, which can be provided from a variety of sources, not simply from persistence (this forcing rarely comes from persistence, most commonly it comes from large models, or from climatological forecasts)

Secondly, as I mentioned in my first review, I feel it is very misleading to refer to the long term use of persistence forcing as 'operational' in any sense here. Having worked in operational regional oceanography I don't think ROMS would ever be used in this way in an operational setting, certainly I have never heard of anything like this. In almost all operational settings, a forecast from a broader model would be available and the model would be forced with this. I don't feel the comparison needs to be re-done, but this caveat needs to properly clarify that it is not being compared to anything resembling a regularly used configuration, and certainly not to anything operational. 'Operational' has a specific meaning in this context and does not seem to be correctly used here - the comments made in the paper imply a very poor approach is taken in operational ocean forecasting, which is not the case. Perhaps there is some confusion over the terminology here. In operational cases (and I would imagine almost all research cases) ROMS would not be used over these timescales with persistence forcing, far better options exist and would be used.

I would like to see this updated, with any reference to operational applications using persistence removed. I suggest something along the lines of the following:

'*The comparisons between OceanNet and ROMS can also be considered to have a major caveat: ROMS as a regional ocean model depends on **providing** forcing conditions at open boundaries, for which persistent boundaries were provided in this study. **This method would not be used in meaningful prediction scenarios over the timescales considered here, and as such** it may be more fair to compare the performance of OceanNet to a model that does not require boundary conditions, such as a global ocean model**, or to a configuration of ROMS forced with boundary conditions taken from predictions produced by a global ocean model."***

The authors also do not seem to have made any reference in these caveats to the use of persistence forcing for the atmospheric forcing conditions as I flagged as a concern in my first review. Again this element of the set up severely degrades the quality of the predictions from ROMS, and again ROMS would not be used to provide forecasts over these timescales with persistent atmospheric forcing. I think this caveat must be noted in the paper.

We appreciate the reviewer for the thoughtful comments and assistance in revision the texts. We have adopted the suggested revision to amend the statement about persistence surface and boundary forcing:

*"The comparisons between OceanNet and ROMS can also be considered to have a major caveat: ROMS as a regional ocean model depends on providing forcing conditions on the ocean surface and at open boundaries, for which persistence was provided in this study. This method would not be used in conventional prediction scenarios over the timescales considered here, and as such it may be more fair to compare the performance of OceanNet to a model that does not require boundary conditions, such as a*

*global ocean model, or to a configuration of ROMS forced with forcing and boundary conditions taken from predictions produced by a global model.***"**

Following my review, the authors updated the paper to state: *"Currently, almost all global ocean and atmosphere forecasts extend only 7-10 days".* This is not correct, many operational forecast centres run forecasts for almost all lead times. Short term systems give forecasts for 7-10 days, but many forecast centres run operational sub-seasonal to seasonal (S2S) forecast systems (generally providing ocean and atmospheric forecasts within a coupled system) which predict out to a few months, and many other systems provide even longer forecasts, all the way out to mutli-decadal climate forecasts and beyond. The paper they cite (Jacox et al., 2020) seems to be a *research* application of marine biology predictions, not an operational system, and not forecasting the ocean itself. The incorrect statements over *operational* forecasts must be amended.

Similarly, the paper now states *"in practice. Persistence is commonly used in short- to medium-term ocean forecasting due to its simplicity (e.g., Jacox et (2020)), but it does not account for changes in climatic conditions such as those driven by El Niño or other large-scale climate."* Persistence is not 'commonly' used, and the issues with persistence are far greater than this - persistence forecasts do not account for *any* variability, over any timescales. For applications such as those in the paper it's not just the lack of climatic variability, but any weather impacts, such as winds, storms/hurricanes, warming or cooling over a few days/weeks etc. The caveat here does not accurately capture the significant limitations of the use of persistence.

We thank the reviewer for the thoughtful comments. Following your guidance, we have amended the paragraph as the following:

"*In regional ocean forecasting, defining surface and boundary forcing is a significant challenge, particularly when accurate and continuous global ocean and atmosphere forecasting data for extended periods are unavailable. In this study, persistence refers to the assumption that future conditions will resemble past conditions. Persistence is sometimes used in short- to medium-term ocean forecasting due to its simplicity (e.g., Jacox et al. 2020), but it does not account for changes in weather and climate conditions. While persistence can provide a baseline, it is not expected to capture the full variability or trends in long-term forecasts. We acknowledge the limitations of using persistent forcing to drive ROMS forecasts in this study.*"

In general, I think that paper has some really interesting science in it, but still needs notable work in describing the methods, and in being clear about the caveats of the comparison given. Along with clarification over the alternative methods to iterate/time-step the models used.

We thank the reviewer again for her very thoughtful and constructive comments and suggestions, which have greatly improved the quality of this manuscript.

---

## Author Response (AR3)

**3rd Review of Long-term Prediction of the Gulf Stream Meander Using OceanNet: a Principled Neural Operator-based Digital Twin**

Many thanks for the clarifications, particularly around the model. I want to reiterate that I think this is interesting work, and would love to see it published, however I maintain my view that some amendments are needed first.

We sincerely appreciate the time and effort both the editor and the reviewer have dedicated to reviewing our manuscript and providing your valuable feedback. We are grateful for your insightful comments and valuable suggestions, all of which have been carefully incorporated in our revision.

The improvements to the model description have hugely improved readability. Thank you for this. However, the use of the convolutional layers are still not mentioned here, please could they be added (the figure 3 caption now includes reference to these, a similar sentence in the model text would help considerably). I note the authors comment that the specifics of the FNO are described elsewhere, and agree that a full and thorough model design paper should not form part of this paper. However, I feel as it stands the text gives a brief description of some parts of the model, ignoring others... a full (but very brief) description of the model is needed – as a description of only part leaves the reader assuming that the model only consists of the parts mentioned, and is misleading. To be clear, I am requesting only a brief overview, but which includes all elements of the model to a simple degree. Currently the convolutional layers are entirely missed in the text.

The Figure 3 caption has been updated and the paragraph starting at line 158 has been changed to reflect the requested brief description of the model.

Re figure 3, I note the authors reluctance to make changes to the figure due to a sister paper. However, in my view, its current form is misleading, and incredibly unhelpful to the reader. The location of the start and end of the curly braces between part a and b of the figure  very much seem to be incorrect. Leaving this misleading element (even while acknowledging this is present in the sister paper) is, to me, incredibly unhelpful to readers. The curly braces currently sit part way through the convolution layers, implying that some of the convolutions belong to the Fourier layer, and some don't. My understanding is that the curly braces should begin and end at the edges of the yellow 'Fourier layer' box. I would like to see the curly braces correctly located, so the start and end clearly define what is included in part b of the figure.

Further to this, it is still unclear to me which part of the figure is included in the definition of $N$. If this is the output from part a, can this be clarified. To me, this is now even more confusing

than before --- the caption implies that N refers just to the Fourier part ("(b) The Fourier Neural Operator, depicted as N."), but part c of the figure 3 shows that N is applied to x... if both these are correct, then when/how are the convolutions in fig3a applied to x?

I think the authors would benefit from having a discussion with someone who was not involved in the creation of the figure, to try and ensure the figure, caption and text make sense to someone who has no prior knowledge of the figure. I am able to offer my time if the authors would like to discuss this by v/c, but it may be just a productive to speak to a colleague in their group/deparment who was not involved with the original figure creation.

Figure 3 and its caption have been amended. $\mathcal{N}$ is correctly referred to as the Fourier Neural Operator. The OceanNet model is comprised of a Fourier Neural Operator, a numerical integration scheme, and a loss function. As conveyed by the mathematics, the Fourier Neural Operator, $\mathcal{N}$, predicts the increment of the system while the numerical integration scheme, **H**, uses the initial field **X** and the increment to give the predicted field. In Figure 3, (a) shows a high-level overview of the Fourier Neural Operator, (b) shows a high-level overview of the Fourier Layers contained in the Fourier Neural Operator. $\mathcal{N}$ is also referred to as the neural network in the sections containing the mathematical theory of the model since we are generalizing (e.g. it could be any neural network).

For me the added lines 181-182 "In practical terms, a future timestep X(t + Δt) is predicted by feeding the initial image X(t) into our neural network N with parameters θ. The numerical integration scheme H is then applied to the outputs as discussed in Chattopadhyay and Hassanzadeh (2023)." do not make sense with other comments in the paper and in response to the review. In the majority of cases (and that discussed here), the tendency, Δt, is predicted by N. Then H is applied to give X(t + Δt). This would read better/make more sense with something like:

"In practical terms, the increment, Δt, is obtained by feeding the initial image X(t) into our neural network N with parameters θ. The numerical integration scheme H is then applied to the this as discussed in Chattopadhyay and Hassanzadeh (2023) to give the future timestep X(t + Δt)"

Thank you for the close attention to detail. This mistake has been corrected as recommended.

The added description in section 2.3.2 hugely improves my understanding of the methods, many thanks. Its particularly useful for enlightening me on what is meant by 'no integration scheme'. I still however feel that the phrase 'no integration scheme' is a bit confusing. Please could this be changed to "directly predicting future fields". I feel it's important to clarify that these experiments have not simply removed the way the increment is applied, but instead

retrained a new model, N, to do a totally different thing. 'no integration scheme' does not capture this message. Please could this be changed throughout the text and figure captions.

This information on (what are now) lines 204-207 has been amended:
"In Sect. \ref{Results}, experiments are described for a variety of models, some of which did not employ a numerical integration scheme in their methods. In such cases, the integration scheme is not present during the training of the neural network, $\mathcal{N}$, thus the model is directly predicting the next timestep as is commonly seen in CNN and U-NET models such as those discussed in Sect. \ref{sect:DLOP}. The equation representing such models can be given as:..."

Furthermore, a recall to this method has been added to lines 306 and 307, where the references to models without an integration scheme begin:
"The RMSE, anomaly correlation coefficient (ACC), and MHD are compared across different iterations of the DLOP and FNO models, focusing on integration schemes and loss function terms. The two integration schemes compared were the absence of integration (Eq. \ref{noint}) and PEC. Recall that a difference in integration schemes corresponds to an entire retraining of the model and thus results in a different model."

The description of v(t) in figure 3 caption has hugely improved readability, thanks. However, it seems the tilda is lost on some of the references – some should refer to v_tilda, not v.

Thank you for catching this error. The typo has been corrected.

I note the authors response to my comments re the inputs of H. I maintain my viewpoint that H should take both x and N as inputs. The authors note 'While our selection of integration scheme is implicit and thus depends on both X(t) and N[X(t), \theta], this does not have to be the case in general.' In my view, the *general approach* would be to have both variables available as inputs, in some cases both are used, in other cases only one. I.e. F(x,y), sometimes F(x,y)=x+y, sometimes F(x,y)=x+2. It's not ideal to have a function which takes different inputs, but certainly including all potential inputs provides this generality. By listing h as a function *only* of N, it is then incorrect to give any inputs other than N, and the generality is lost.

This suggestion has been implemented. Where appropriate, H has been given inputs of X(t) and N[X(t), \theta]

The caveats in the use of persistence etc are much better stated now, thank you. However, I still have issue with the line "Persistence is sometimes used in short-to-medium term ocean forecasting due to its simplicity (e.g. Jacox et al. (2020)), but it does not account for changes in weather and climate conditions." I have not thoroughly read the Jacox paper, but firstly, and most importantly it refers to prediction of marine ecosystems. Ocean forecasting (including in the applications of this paper) is very distinct biogeochemical/ecosystem forecasting. Further, this paper seems to be a review paper discussing the predictability of systems, and commenting on how _anomaly_ persistence _contibutes_ to predictability. Again this is very different to stating persistence can be suitable used for forcing. I would very much like to see this single sentence removed.

The sentence has been removed as requested.

Thank you again for the great attention to detail from both the reviewer and the editor- this has greatly improved our submission and we look forward to any further discussions.